# Accelerated Training via Principled Methods for Incrementally Growing Neural Networks

## Abstract

We develop an approach to efficiently grow neural networks, within which parameterization and optimization strategies are designed by considering their effects on the training dynamics. Unlike existing growing methods, which follow simple replication heuristics or utilize auxiliary gradient-based local optimization, we craft a parameterization scheme which dynamically stabilizes weight, activation, and gradient scaling as the architecture evolves, and maintains the inference functionality of the network. To address the optimization difficulty resulting from imbalanced training effort distributed to subnetworks fading in at different growth phases, we propose a learning rate adaption mechanism that rebalances the gradient contribution of these separate subcomponents. Experimental results show that our method achieves comparable or better accuracy than training large fixed-size models, while saving a substantial portion of the original computation budget for training. We demonstrate that these gains translate into real wall-clock training speedups.

## 1 Introduction

Modern neural network design typically follows a "larger is better" rule of thumb, with models consisting of millions of parameters achieving impressive generalization performance across many tasks, including image classification (Krizhevsky et al., 2012; Simonyan & Zisserman, 2015; Real et al., 2019; Zhai et al., 2022), object detection (Girshick, 2015; Liu et al., 2016; Ghiasi et al., 2019), semantic segmentation (Long et al., 2015; Chen et al., 2017; Liu et al., 2019a) and machine translation (Vaswani et al., 2017; Devlin et al., 2019). Within a class of network architecture, deeper or wider variants of a base model typically yield further improvements to accuracy. Residual networks (ResNets) (He et al., 2016b) and wide residual networks (Zagoruyko & Komodakis, 2016) illustrate this trend in convolutional neural network (CNN) architectures. Dramatically scaling up network size into the billions of parameter regime has recently revolutionized transformer-based language modeling (Vaswani et al., 2017; Devlin et al., 2019; Brown et al., 2020).

The size of these models imposes prohibitive training costs and motivate techniques that offer cheaper alternatives to select and deploy networks. For example, hyperparameter tuning is notoriously expensive as it commonly relies on training the network multiple times, and recent techniques aim to circumvent this by making hyperparameters transferable between models of different sizes, allowing them to be tuned on a small network prior to training the original model once (Yang et al., 2021).

Our approach incorporates these ideas, but extends the scope of transferability to include the parameters of the model itself. Rather than view training small and large models as separate events, we grow a small model into a large one through many intermediate steps, each of which introduces additional parameters to the network. Our contribution is to do so in a manner that preserves the function computed by the model at each growth step (functional continuity) and offers stable training dynamics, while also saving compute by leveraging intermediate solutions. More specifically, we use partially trained subnetworks as scaffolding that accelerates training of newly added parameters, yielding greater overall efficiency than training a large static model from scratch.

Motivating this general strategy, we view aspects of prior works as hinting that deep network training may naturally be amenable to dynamically growing model size. For example, residual connections (He et al., 2016b) introduce depth-wise shortcuts, solving a gradient vanishing issue and thereby making very deep networks end-to-end trainable. Prior to ResNet, manually circumventing this issue involved

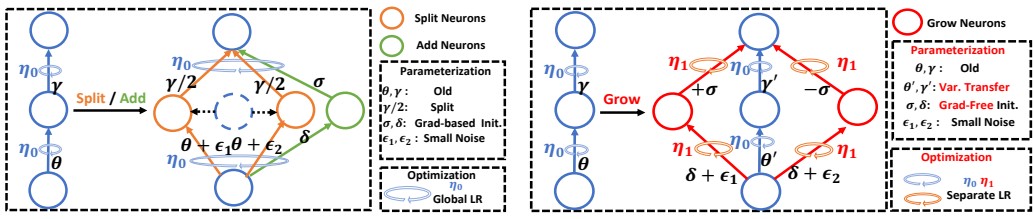

(a) Existing Methods: Splitting Init, Global LR  (b) Ours: Function-Preserving Init, Stagewise LR

Figure 1: Dynamic network growth strategies. Different from (a) which reply on either splitting (Chen et al., 2016; Liu et al., 2019b; Wu et al., 2020b) or adding neurons with auxiliary local optimization (Wu et al., 2020a; Evci et al., 2022), our initialization (b) of new neurons is random but function-preserving. Additionally, our separate learning rate (LR) scheduler governs weight updating in order to address the discrepancy in total accumulated training between different growth stages.

adding outputs and losses to intermediate layers (Szegedy et al., 2015), effectively causing a shallower subnetwork to train first and bootstrap the training of the full network. Larsson et al. (2016) show that end-to-end training of FractalNet, an alternative shortcut architecture, implicitly trains shallower subnetworks first. If such phenomena, though perhaps difficult to analyze, occurs more broadly, it suggests that one might achieve computational advantage by adopting an explicit growth strategy that matches the implicit subnetwork training schedule which occurs within a large static network.

While overparameterization benefits generalization, a more detailed view suggests possible compatibility between the desire to maintain an overparameterized deep network and to dynamically grow such a network. The "double-descent" bias-variance trade-off curve (Belkin et al., 2019) indicates that large model capacity may be a safe strategy to ensure operation in the modern interpolating regime with consequently low test error. Small models, as we take for a starting point in dynamic growth, might not be sufficiently overparameterized and incur higher test error. However, Nakkiran et al. (2020) experimentally observe double-descent occurs with respect to both model size and number of training epochs. To remain in the interpolating regime, a model must be overparameterized relative to the amount it has been trained, which can be satisfied by an appropriate growth schedule.

Competing recent efforts to grow deep models from simple architectures (Chen et al., 2016; Li et al., 2019; Dai et al., 2019; Liu et al., 2019b; Wu et al., 2020b; Wen et al., 2020; Wu et al., 2020a; Yuan et al., 2021; Evci et al., 2022) draw inspiration from other sources, such as the progressive development processes of biological brains. In particular, Net2Net (Chen et al., 2016) grows the network by randomly splitting learned neurons from previous phases. This replication scheme, shown in Figure 1(a) is a common paradigm for most existing methods. Splitting steepest descent (Wu et al., 2020b) determines which neurons to split and how to split them by solving a combinatorial optimization problem with auxiliary variables. Firefly (Wu et al., 2020a) further improves flexibility by incorporating optimization for adding new neurons. Both methods outperform simple heuristics, but require additional training effort in their gradient-based parameterization schemes. Furthermore, all existing methods use a global learning rate scheduler to govern weight updates, ignoring the discrepancy in total training time among subnetworks introduced in different growth phases.

We develop a growing framework around the principles of enforcing transferability of parameter settings from smaller to larger models (extending Yang et al. (2021)), offering functional continuity, smoothing optimization dynamics, and rebalancing learning rates between older and newer subnetworks. Figure 1(b) illustrates key differences with prior work. Our core contributions are:

- **Parameterization using Variance Transfer:** We propose a parameterization scheme accounting for the variance transition among networks of smaller and larger width in a single training process. Initialization of new weights is gradient-free and requires neither additional memory nor training.
- **Improved Optimization with Rate Adaptation:** Subnetworks trained for different lengths have distinct learning rate schedules, with dynamic relative scaling driven by weight norm statistics.
- **Better Performance and Broad Applicability:** Our method not only trains networks fast, but also yields excellent generalization accuracy, even outperforming the original fixed-size models. Flexibility in designing a network growth curve allows choosing different trade-offs between training resources and accuracy. Furthermore, adopting an adaptive batch size schedule provides acceleration in terms of wall-clock training time. We demonstrate results on image classification and machine translation tasks, across a diverse set of network architectures.

## 2   RELATED WORK

**Network Growing.** A diverse range of techniques train models by progressively expanding the network architecture (Wei et al., 2016; Elsken et al., 2018; Dai et al., 2019; Wen et al., 2020; Yuan et al., 2021). Within this space, the methods of Chen et al. (2016); Liu et al. (2019b); Wu et al. (2020b;a); Evci et al. (2022) are most relevant to our focus – incrementally growing network width across multiple training stages. Net2Net (Chen et al., 2016) proposes a gradient-free neuron splitting scheme via replication, enabling knowledge transfer from previous training phases; initialization of new weights follows simple heuristics. Liu et al. (2019b)'s Splitting approach derives a gradient-based scheme for duplicating neurons by formulating a combinatorial optimization problem. FireFly (Wu et al., 2020a) gains flexibility by also incorporating brand new neurons. Both of these methods improve Net2Net's initialization scheme by solving an optimization problem with auxiliary variables, at the cost of extra training effort. GradMax (Evci et al., 2022), in consideration of training dynamics, performs initialization via solving a singular value decomposition (SVD) problem.

**Neural Architecture Search (NAS) and Pruning.** Another subset of methods mix growth with dynamic reconfiguration aimed at discovering or pruning task-optimized architectures. Network Morphism (Wei et al., 2016) searches for efficient networks by extending layers while preserving the parameters. Autogrow (Wen et al., 2020) takes an AutoML approach governed by heuristic growing and stopping policies. Yuan et al. (2021) combine learned pruning with a sampling strategy that dynamically increases or decreases network size. Unlike these methods, we focus on the mechanics of growth when the target architecture is known, addressing the question of how to best transition weight and optimizer state to continue training an incrementally larger model. NAS and pruning are orthogonal to, though potentially compatible with, the technical approach we develop.

**Hyperparameter Transfer.** Yogatama & Mann (2014); Perrone et al. (2018); Horváth et al. (2021) explore transferrable hyperparameter (HP) tuning. The recent Tensor Program (TP) work of Yang & Hu (2021); Yang et al. (2021) focuses on zero-shot HP transfer across model scale and establishes a principled network parameterization scheme to facilitate HP transfer. This serves as an anchor for our strategy, though, as Section 3 details, modifications are required to account for dynamic growth.

**Learning Rate Adaptation.** Surprisingly, the existing spectrum of network growing techniques utilize relatively standard learning rate schedules and do not address potential discrepancy among subcomponents added at different phases. One might expect that newer weights should have higher learning rates than older weights. While general-purpose adaptive optimizers (*e.g.*, AdaGrad (Duchi et al., 2011), RMSProp (Tieleman et al., 2012), Adam (Kingma & Ba, 2015), AvaGrad (Savarese et al., 2021)) might ameliorate this issue, we choose to explicitly account for the discrepancy. As layer-adaptive learning rates (LARS) (Ginsburg et al., 2018; You et al., 2020) benefit in some contents, we explore further learning rate adaption specific to both layer and growth stage.

## 3   METHOD

### 3.1   PARAMETERIZATION AND OPTIMIZATION WITH GROWING DYNAMICS

**Functionality Preservation**. We grow network capacity by expanding the width of computational units (*e.g.,* hidden dimensions in linear layers, filters in convolutional layers). To illustrate our scheme, consider a 3-layer fully-connected network with ReLU activations $\phi$:

$$\boldsymbol{h}^{in} = \phi(\boldsymbol{W}^{in}\boldsymbol{x}), \quad \boldsymbol{h}^o = \phi(\boldsymbol{W}^h\boldsymbol{h}^{in}), \quad \boldsymbol{y} = \boldsymbol{W}^{out}\boldsymbol{h}^o, \tag{1}$$

where $\boldsymbol{x} \in \mathbb{R}^{C_{in}}$ is the network input, $\boldsymbol{y} \in \mathbb{R}^{C_{out}}$ is the output, and $\boldsymbol{h}^{in} \in \mathbb{R}^{H_{in}}, \boldsymbol{h}^o \in \mathbb{R}^{H_{out}}$ are the hidden activations. In this case, $\boldsymbol{W}^{in}$ is a $H_{in} \times C_{in}$ matrix, while $\boldsymbol{W}^h$ is $H_{out} \times H_{in}$ and $\boldsymbol{W}^{out}$ is $C_{out} \times H_{out}$. After training the network for a few epochs, we increase its capacity by increasing the dimensionality of each hidden state, *i.e.,* from $H_{in}$ and $H_{out}$ to $\widehat{H_{in}}$ and $\widehat{H_{out}}$, respectively. The layer parameter matrices $\boldsymbol{W}$ have their shapes changed accordingly and become $\widehat{\boldsymbol{W}}$.

Figure 2 illustrates the process for initializing $\widehat{\boldsymbol{W}}$.[1] As Figure 2(a) shows, we first expand $\boldsymbol{W}^{in}$ along the output dimension by adding two copies of new weights $\boldsymbol{W}_n^{in}$ of shape $\frac{\widehat{H_{in}}-H_{in}}{2} \times C_{in}$,

---

[1] We defer the transformation between $\boldsymbol{W}_{old}$ and $\boldsymbol{W}_{new}$ to the next subsection. It involves rescaling by constant factors, does not affect network functionality, and is omitted in Eq. 1- 4 for simplicity.

Table 1: Parameterization and optimization transition for different layers during growing.

|  |  | Input Layer | Hidden Layer | Output Layer |
|---|---|---|---|---|
| Init. | Old Weighs Scaling | $1$ | $\sqrt{H_{in}/\widehat{H_{in}}}$ | $H_{out}/\widehat{H_{out}}$ |
|  | New Weighs Init. | $1/C_{in}$ | $1/\widehat{H_{in}}$ | $1/(\widehat{H_{out}})^2$ |
| LR Adapt. | 0-th Stage | $1$ | $1$ | $1/H_{out}^0$ |
|  | i-th Stage | $\frac{\|\boldsymbol{W}_i^{in}\setminus\boldsymbol{W}_{i-1}^{in}\|}{\|\boldsymbol{W}_0^{in}\|}$ | $\frac{\|\boldsymbol{W}_i^h\setminus\boldsymbol{W}_{i-1}^h\|}{\|\boldsymbol{W}_0^h\|}$ | $\frac{\|\boldsymbol{W}_i^{out}\setminus\boldsymbol{W}_{i-1}^{out}\|}{\|\boldsymbol{W}_0^{out}\|}$ |

generating new features $\boldsymbol{h}_n^{in} = \phi(\boldsymbol{W}_n^{in}\boldsymbol{x})$ and changing the first set of activations from $\boldsymbol{h}^{in}$ to

$$\widehat{\boldsymbol{h}^{in}} = concat(\boldsymbol{h}^{in}, \boldsymbol{h}_n^{in}, \boldsymbol{h}_n^{in}). \tag{2}$$

Next, we expand $\boldsymbol{W}^h$ across both input and output dimensions, as shown in Figure 2(b). We initialize new weights $\boldsymbol{W}_e^h$ of shape $H_{out} \times \frac{\widehat{H_{in}}-H_{in}}{2}$ and add to $\boldsymbol{W}^h$ two copies of it with different signs: $+\boldsymbol{W}_e^h$ and $-\boldsymbol{W}_e^h$. This preserves the output of the layer since

$$\phi(\boldsymbol{W}^h\boldsymbol{h}^{in} + \boldsymbol{W}_e^h\boldsymbol{h}_n^{in} + (-\boldsymbol{W}_e)\boldsymbol{h}_n^{in}) = \phi(\boldsymbol{W}^h\boldsymbol{h}^{in}) = \boldsymbol{h}^o.$$

We then add two copies of new weights $\boldsymbol{W}_n^h$, which has shape $\frac{\widehat{H_{out}}-H_{out}}{2} \times \widehat{H_{in}}$, yielding activations

$$\widehat{\boldsymbol{h}^o} = concat(\boldsymbol{h}^o, \phi(\boldsymbol{W}_n^h\widehat{\boldsymbol{h}^{in}}), \phi(\boldsymbol{W}_n^h\widehat{\boldsymbol{h}^{in}})). \tag{3}$$

We similarity expand $\boldsymbol{W}^{out}$ to match the dimension of $\widehat{\boldsymbol{h}_o}$. As Figure 2(c) shows, the final output is:

$$\widehat{\boldsymbol{y}} = \phi(\boldsymbol{W}^{out}\boldsymbol{h}^o + \boldsymbol{W}_e^{out}\phi(\boldsymbol{W}_n^h\widehat{\boldsymbol{h}^{in}}) + (-\boldsymbol{W}_e^{out})\phi(\boldsymbol{W}_n^h\widehat{\boldsymbol{h}^{in}}) = \boldsymbol{y} \tag{4}$$

which preserves the original output features in Eq. 1.

**Weights Initialization with Variance Transfer (VT).** Yang et al. (2021) investigate weight scaling with width at initialization, allowing hyperparameter transfer by calibrating variance across model size. They modify the variance of output layer weights from the commonly used $1/\text{fan}_{in}$ to $1/\text{fan}_{in}^2$. We adopt this same correction for the added weights with new width: $\boldsymbol{W}^{out}$ and $\boldsymbol{W}_e^{out}$ are initialized with variances of $1/{H_{out}}^2$ and $1/\widehat{H_{out}}^2$, respectively.

However, this correction considers training differently-sized models separately, which fails to accommodate the training dynamics in which width grows incrementally. To make the weights of the old subnetwork $\boldsymbol{W}_{old}^{out} \sim \mathcal{N}(0, 1/(H_{out})^2)$ compatible with the entire weight tensor parameterization, we rescale it with the $\text{fan}_{in}$ ratio as: $\boldsymbol{W}_{new}^{out} = \boldsymbol{W}_{old}^{out} \cdot H_{out}/\widehat{H_{out}}$. Also see Table 1 (top).

This parameterization rule transfers to modern convolutional networks with batch normalization (BN). Given a weight scaling ratio of $c$, the running mean $\mu$ and variance $\sigma$ of BN layers are modified as $c\mu$ and $c^2\sigma$, respectively.

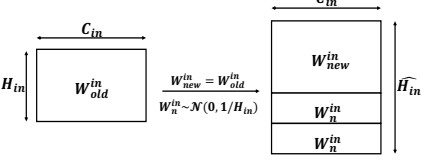

(a) Input Layer

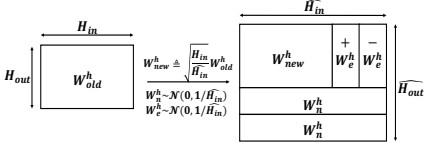

(b) Hidden Layer

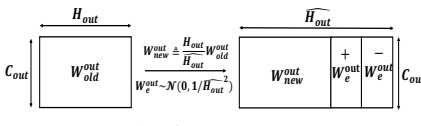

(c) Output Layer

Figure 2: Initialization scheme. In practice, we also add noise to the expanded parameter sets for symmetry breaking.

**Stage-wise Learning Rate Adaptation (RA).** Following (Yang et al., 2021), we employ a learning rate scaling factor of $\propto 1/\text{fan}_{in}$ on the output layer when using SGD, compensating for the initialization scheme. However, subnetworks from different growth stages still share a global learning rate, though they have trained for different lengths. This may cause divergent behavior among the corresponding weights, making the training iterations after growing sensitive to the scale of the newly-initialized weights. Instead of adjusting newly added parameters via local optimization (Wu et al., 2020a; Evci et al., 2022), we govern the update of each subnetwork in a stage-wise manner.

Suppose a layer scales up with width at different growing stages $S_0 \subset S_1 \subset ... \subset S_{N-1}$ with associated weight and gradient tensors as $\boldsymbol{W}_0 \subset \boldsymbol{W}_1 \subset ... \subset \boldsymbol{W}_{N-1}$ and $\boldsymbol{G}_0 \subset \boldsymbol{G}_1 \subset ... \subset \boldsymbol{G}_{N-1}$, respectively. We adapt the learning rate and update the $i$-th sub-weights $\boldsymbol{W}_i \setminus \boldsymbol{W}_{i-1}$ as:

$$\eta_i = \eta_0 * \frac{f(S_i \setminus S_{i-1})}{f(S_0)}, \quad \boldsymbol{W}_i \setminus \boldsymbol{W}_{i-1} \leftarrow -\eta_i * (\boldsymbol{G}_i \setminus \boldsymbol{G}_{i-1}), \quad i > 0 \tag{5}$$

where $\eta_0$ is the base learning rate and $f$ is an implicit function that maps subnetworks of different stages to corresponding train-time statistics. Table 1 (bottom) summarizes our LR adaptation rule for SGD when $f$ is instantiated as weight norm. Alternative heuristics are possible; see Appendix A.1.

## 3.2 FLEXIBLE AND EFFICIENT GROWTH SCHEDULER

We train the model for $T_{total}$ epochs by expanding the channel number of each layer to $C_{final}$ across $N$ growth phases. Existing methods (Liu et al., 2019b; Wu et al., 2020a) fail to derive a systemic way for distributing training resources across a growth trajectory. Toward maximizing efficiency, we experiment with a coupling between model size and training epoch allocation.

**Architectural Scheduler.** We denote initial channel width as $C_0$ and expand exponentially:

$$C_n = \begin{cases} C_{n-1} + \lfloor p_c C_{n-1} \rfloor_2 & \text{if} \quad n < N-1 \\ C_{final} & \text{if} \quad n = N-1 \end{cases} \quad (6)$$

where $\lfloor \cdot \rfloor_2$ rounds to the nearest even number and $p_c$ is the growth rate between stages.

**Epoch Scheduler.** We denote number of epochs assigned to $n$-th training stage as $T_n$, with $\sum_{n=0}^{N-1} T_n = T_{total}$. We similarly adapt $T_n$ via an exponential growing scheduler:

---

**Algorithm 1** : Growing using Var. Transfer and Learning Rate Adapt. with Flexible Scheduler

---

**Input:** Data $\boldsymbol{X}$, labels $\boldsymbol{Y}$, task loss $L$
**Output:** Grown model $S$
Initialize: $S_0$ with $C_0, T_0, B_0, \eta_0$
**for** n = 0 **to** $N-1$ **do**
   **if** $n > 0$ **then**
      Init. $S_n$ from $S_{n-1}$ using VT in Table 1.
      Update $C_n$ and $T_n$ using Eq. 6 and Eq. 7.
      Update $B_n$ using Eq. 8 (optional)
      $\text{Iter}_{total} = T_n * len(X) // B_n$
   **end if**
   **for** i = 1 **to** $\text{Iter}_{total}$ **do**
      Forward and calculate $l = L(S_n(\boldsymbol{x}), \boldsymbol{y})$.
      Back propagation with $l$.
      Update each sub-component using Eq. 5.
   **end for**
**end for**
return $S_{N-1}$

---

$$T_n = \begin{cases} T_{n-1} + \lfloor p_t T_{n-1} \rfloor & \text{if} \quad n < N-1 \\ T_{total} - \sum_{i=0}^{N-2} T_i & \text{if} \quad n = N-1 \end{cases} \quad (7)$$

**Wall-clock Speedup via Batch Size Adaptation.** Though the smaller architectures in early growth stages require fewer FLOPs, hardware capabilities may still restrict practical gains. When growing width, in order to ensure that small models fully utilize the benefits of GPU parallelism, we adapt the batch size along with the exponentially-growing architecture in a reverse order:

$$B_{n-1} = \begin{cases} B_{base} & \text{if} \quad n = N \\ B_n + \lfloor p_b B_n \rfloor & \text{if} \quad n < N \end{cases} \quad (8)$$

where $B_{base}$ is the batch size of the large baseline model. Algorithm 1 summarizes our full method.

## 4 EXPERIMENTS

We evaluate on image classification and machine translation tasks. For image classification, we use CIFAR-10 (Krizhevsky et al., 2014), CIFAR-100 (Krizhevsky et al., 2014) and ImageNet (Deng et al., 2009). For the neural machine translation, we use the IWSLT'14 dataset (Cettolo et al., 2014) and report the BLEU (Papineni et al., 2002) score on German to English (De-En) translation task.

**Large Baselines via Fixed-size Training.** We use VGG-11 (Simonyan & Zisserman, 2015) with BatchNorm (Ioffe & Szegedy, 2015), ResNet-20 (He et al., 2016a), MobileNetV1 (Howard et al., 2017) for CIFAR-10 and VGG-19 with BatchNorm, ResNet-18, MobileNetV1 for CIFAR-100. We follow Huang et al. (2016) for data augmentation and processing, adopting random shifts/mirroring and channel-wise normalization. CIFAR-10 and CIFAR-100 models are trained for 160 and 200 epochs respectively, with a batch size of 128 and initial learning rate (LR) of 0.1 using SGD. We adopt a cosine LR schedule and set the weights decay and momentum as 5e-4 and 0.9. For ImageNet, we train the baseline ResNet-50 and MobileNetV1 (Howard et al., 2017) using SGD with batch sizes of 256 and 512, respectively. We adopt the same data augmentation scheme as Gross & Wilber (2016), the cosine LR scheduler with initial LR of 0.1, weight decay of 1e-4 and momentum of 0.9.

Table 2: Growing ResNet-20, VGG-11, and MobileNetV1 on CIFAR-10.

| Method | ResNet-20 | | VGG-11 | | MobileNetv1 | |
|---|---|---|---|---|---|---|
| | Train Cost(%) ↓ | Test Accuracy(%) ↑ | Train Cost(%) ↓ | Test Accuracy(%) ↑ | Train Cost(%) ↓ | Test Accuracy(%) ↑ |
| Large Baseline | 100 | $92.62 \pm 0.15$ | 100 | $92.14 \pm 0.22$ | 100 | $92.27 \pm 0.11$ |
| Net2Net | **54.90** | $91.60 \pm 0.21$ | **52.91** | $91.78 \pm 0.27$ | **53.80** | $90.34 \pm 0.20$ |
| Splitting | 70.69 | $91.80 \pm 0.10$ | 63.76 | $91.88 \pm 0.15$ | 65.92 | $91.50 \pm 0.06$ |
| FireFly-split | 58.47 | $91.78 \pm 0.11$ | 56.18 | $91.91 \pm 0.15$ | 56.37 | $91.56 \pm 0.06$ |
| FireFly | 68.96 | $92.10 \pm 0.13$ | 60.24 | $92.08 \pm 0.16$ | 62.12 | $91.69 \pm 0.07$ |
| Ours | **54.90** | **$92.53 \pm 0.11$** | **52.91** | **$92.34 \pm 0.15$** | **53.80** | **$92.01 \pm 0.10$** |

Table 3: Growing ResNet-18, VGG-19, and MobileNetV1 on CIFAR-100.

| Method | ResNet-18 | | VGG-19 | | MobileNetv1 | |
|---|---|---|---|---|---|---|
| | Train Cost(%) ↓ | Test Accuracy(%) ↑ | Train Cost(%) ↓ | Test Accuracy(%) ↑ | Train Cost(%) ↓ | Test Accuracy(%) ↑ |
| Large Baseline | 100 | $78.36 \pm 0.12$ | 100 | $72.59 \pm 0.23$ | 100 | $72.13 \pm 0.13$ |
| Net2Net | **52.63** | $76.48 \pm 0.20$ | **52.08** | $71.88 \pm 0.24$ | **52.90** | $70.01 \pm 0.20$ |
| Splitting | 68.01 | $77.01 \pm 0.12$ | 60.12 | $71.96 \pm 0.12$ | 58.39 | $70.45 \pm 0.10$ |
| FireFly-split | 56.11 | $77.22 \pm 0.11$ | 54.64 | $72.19 \pm 0.14$ | 54.36 | $70.69 \pm 0.11$ |
| FireFly | 65.77 | $77.25 \pm 0.12$ | 57.48 | $72.79 \pm 0.13$ | 56.49 | $70.99 \pm 0.10$ |
| Ours | **52.63** | **$78.12 \pm 0.15$** | **52.08** | **$73.26 \pm 0.14$** | **52.90** | **$71.53 \pm 0.13$** |

For IWSLT'14, we train an Encoder-Decoder Transformer (6 attention blocks each) (Vaswani et al., 2017). We set width as $d_{model} = 1/4 d_{ffn} = 512$, the number of heads $n_{head} = 8$ and $d_k = d_q = d_v = d_{model}/n_{head} = 64$. We train the model using Adam for 20 epochs with learning rate 1e-3 and $(\beta_1, \beta_2) = (0.9, 0.98)$. Batch size is 1500 and we use 4000 warm up iterations.

**Implementation Details.** We compare with the growing methods Net2Net (Chen et al., 2016), Splitting (Liu et al., 2019b), FireFly-split, FireFly (Wu et al., 2020a) and GradMax (Evci et al., 2022).

For image classification, we run the comparison methods except GradMax alongside our algorithm for all architectures under the same growing scheduler. For the architecture scheduler, we set $p_c$ as 0.2 and $C_0$ as 1/4 of large baselines in Eq. 6 for all layers and grow the seed architecture within $N = 9$ stages towards the large ones. For epoch scheduler, we set $p_t$ as 0.2, $T_0$ as 8, 10, and 4 in Eq. 7 on CIAFR-10, CIFAR-100, and ImageNet respectively. Total training epochs $T_{total}$ are the same as the respective large fixed-size models. For CIFAR-10 and CIFAR-100, we train the models and report the results averaging over 3 random seeds.

For machine translation, we grow the encoder and decoder layers' widths while fixing the embedding layer dimension for a consistent positional encoding table. The total number of growing stages is 4, each trained for 5 epochs. The initial width is 1/8 of the large baseline (i.e. $d_{model} = 64$ and $d_{k,q,v} = 8$). We set the growing ratio $p_c$ as 1.0 so that $d_{model}$ evolves as 64, 128, 256 and 512.

We train all the models on an NVIDIA 2080Ti 12GB GPU for CIFAR-10, CIFAR-100, and IWSLT'14, and two NVIDIA A40 48GB GPUs for ImageNet.

## 4.1 CIFAR RESULTS

All models grow from a small seed architecture to the full-sized one in 9 stages, each trained for $\{8, 9, 11, 13, 16, 19, 23, 28, 33\}$ epochs (160 total) in CIFAR-10, and $\{10, 12, 14, 17, 20, 24, 29, 35, 39\}$ (200 total) in CIFAR-100. Net2Net follows the design of growing by splitting via simple neuron replication, hence achieving the same training efficiency as our gradient-free method under the same growing schedule. Splitting and Firely require additional training effort for their neuron selection schemes and allocate extra GPU memory for auxiliary variables during the local optimization stage. This is computationally expensive, especially when growing a large model.

**ResNet-20, VGG-11, and MobileNetV1 on CIFAR-10.** Table 2 shows results in terms of test accuracy and training cost calculated based on overall FLOPs. For ResNet-20, Splitting and Firefly

Table 4: ResNet-50 and MobileNetV1 on ImageNet.

| Method | ResNet-50 | | MobileNet-v1 | |
|---|---|---|---|---|
| | Train Cost(%) ↓ | Test Acc.(%) | Train Cost(%) ↓ | Test Acc.(%) |
| Large | 100 | 76.72 | 100 | 70.80 |
| Net2Net | 60.12 | 74.89 | 63.72 | 66.19 |
| FireFly | 71.20 | 75.01 | 86.67 | 66.40 |
| GradMax | - | - | 86.67 | 68.60 |
| Ours | **60.12** | **75.90** | **63.72** | **69.92** |

Table 5: Transformer on IWSLT'14.

| Method | Transformer | |
|---|---|---|
| | Train Cost(%) ↓ | BLEU Score ↑ |
| Large | 100 | 32.84 |
| Net2Net | 64.64 | 30.94 |
| Ours-w/o buffer | 64.64 | 31.42 |
| Ours-w buffer | 64.64 | 31.66 |
| Ours-w buffer-RA | 64.64 | **32.08** |

achieve better test accuracy than Net2Net, which suggest the additional local optimization benefits neuron selection at the cost of training efficiency. Our method requires only $54.9\%$ of the baseline training cost and outperforms all competing methods, while yielding only $0.09p.p$ (percentage points) performance degradation compared to the static baseline. Moreover, our method can even outperform the large fixed-size VGG-11 by $0.20p.p$ test accuracy, while taking only $52.91\%$ of its training cost. For MobileNetV1, our method also achieves the best trade-off between training efficiency and test accuracy among all competitors.

**ResNet-18, VGG-19, and MobileNetV1 on CIFAR-100.** We also evaluate all methods on CIFAR-100 using different network architectures. Results in Table 3 show that Firely consistently achieves better test accuracy than Firefly-split, suggesting that adding new neurons provides more flexibility for exploration than merely splitting. Both Firely and our method achieve better performance than the original VGG-19, suggesting that network growing might have an additional regularizing effect. Our method yields the best accuracy and largest training cost reduction.

## 4.2 IMAGENET RESULTS

We first grow ResNet-50 on ImageNet and compare the performance of our method to Net2Net and FireFly under the same epoch schedule: $\{4, 4, 5, 6, 8, 9, 11, 14, 29\}$ (90 total) with 9 growing phases. We also grow MobileNetV1 from a small seed architecture, which is more challenging than ResNet-50. We train Net2Net and our method using the same scheduler as for ResNet-50. We also compare with Firefly-Opt (a variant of FireFly) and GradMax and report their best results from Evci et al. (2022). Note that both methods not only adopt additional local optimization, but also train with a less efficient growing scheduler: the final full-sized architecture needs to be trained for a $75\%$ fraction while ours only requires $32.2\%$. Table 4 shows that our method dominates all competing approaches.

## 4.3 IWSLT14 DE-EN RESULTS

We grow a Transformer from $d_{model} = 64$ to $d_{model} = 512$ within 4 stages, each trained with 5 epochs using Adam. Applying gradient-based growing methods to the Transformer architecture is non-trivial due to their domain specific design of local optimization. As such, we only compare with Net2Net. We also design variants of our method for self-comparison, based on the adaptation rules for Adam in Appendix A.1. As shown in Table 5, our method generalizes well to the Transformer architecture for the machine translation task. Comparison among variants is also consistent with Table 7, demonstrating the benefit of learning rate adaptation.

## 4.4 ANALYSIS

**Variance Transfer.** We train a simple neural network with 4 convolutional layers on CIFAR-10. The network consists of 4 resolution-preserving convolutional layers; each convolution has 64, 128, 256 and 512 channels, a $3 \times 3$ kernel, and is followed by BatchNorm and ReLU activations. Max-pooling is applied to each layer for a resolution-downsampling of 4, 2, 2, and 2. These CNN layers are then followed by a linear layer for classification. We first alternate this network into four variants, given by combinations of training epochs $\in \{13(1\times), 30(2.3\times)\}$ and initialization methods $\in \{$standard, $\mu$transfer (Yang et al., 2021)$\}$. We also grow from a thin architecture within 3 stages, where the channel number of each layer starts with only 1/4 of the original one, *i.e.,* $\{16, 32, 64, 128\} \rightarrow \{32, 64, 128, 256\} \rightarrow \{64, 128, 256, 512\}$, each stage is trained for 10 epochs.

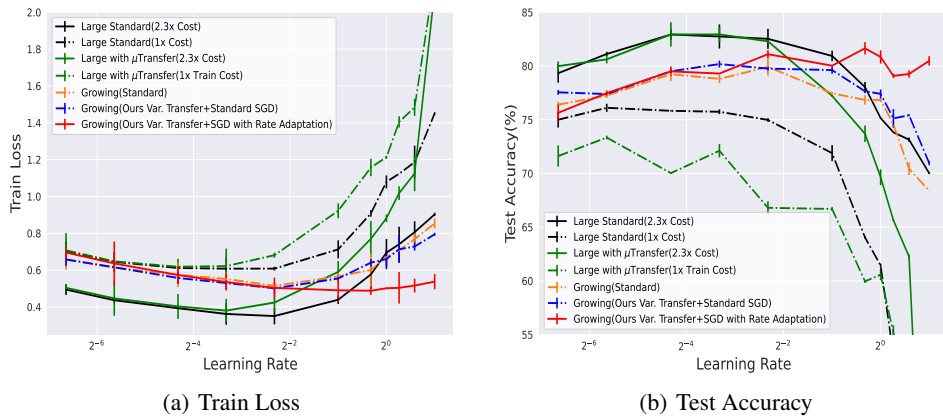

(a) Train Loss

(b) Test Accuracy

Figure 3: Different baselines of 4-layers simple CNN on CIFAR-10.

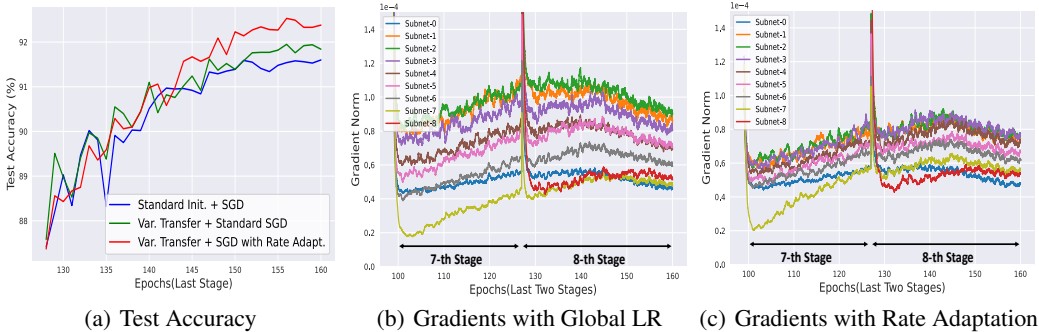

(a) Test Accuracy

(b) Gradients with Global LR

(c) Gradients with Rate Adaptation

Figure 4: (a) Performance with Var. Transfer and Rate Adaptation growing ResNet-20. (b) and (c) visualizes the gradients for different sub-compoents along training in the last two stages.

For network growing, we compare the baselines with standard initialization and variance transfer. We train all baselines using SGD, with weight decay set as 0 and learning rates sweeping over $\{0.01, 0.02, 0.05, 0.1, 0.2, 0.5, 0.8, 1.0, 1.2, 1.5, 2.0\}$. In Figure 3(b), growing with Var. Transfer (blue) achieves overall better test accuracy than standard initialization (orange). Large baselines with merely $\mu$transfer in initialization consistently underperform standard ones, which validate that the compensation from the LR re-scaling is necessary in Yang et al. (2021). We also observe, in both Figure 3(a) and 3(b), all growing baselines outperform fixed-size ones under the same training cost, demonstrating positive regularization effects. We also show the effect of our initialization scheme by comparing test performance on standard ResNet-20 on CIFAR-10. As shown in Figure 4(a), compared with standard initialization, our variance transfer not only achieves better final test accuracy but also appears more stable. See Appendix A.4 for a fully-connected network example.

**Learning Rate Adaptation.** We investigate the value of our proposed stage-wise learning rate adaptation as an optimizer for growing networks. As shown in the red curve in Figure 3, rate adaptation not only bests the train loss and test accuracy among all baselines, but also appears to be more robust over different learning rates. In Figure 4(a), rate adaptation further improves final test accuracy for ResNet-20 on CIFAR-10, under the same initialization scheme.

Figure 4(b) and 4(c) visualize the gradients of different sub-components for the 17-th convolutional layer of ResNet-20 during last two growing phases of standard SGD and rate adaptation, respectively. Our rate adaptation mechanism rebalances subcomponents' gradient contributions to appear in lower divergence than global LR, when components are added at different stages and trained for different durations. In Figure 5, we observe that the LR for newly added Subnet-8 (red) in last stage starts around $1.8\times$ the base LR, then quick adapts to a smoother level. This demonstrates that our method is able to automatically adapt the updates applied to new weights, without any additional local optimization costs (Wu et al., 2020b; Evci et al., 2022). All above show our method has a positive effect in terms of stabilizing training dynamics, which is lost if one attempts to train different subcomponents using a global LR scheduler. Appendix A.2 and A.3 provide more visualizations.

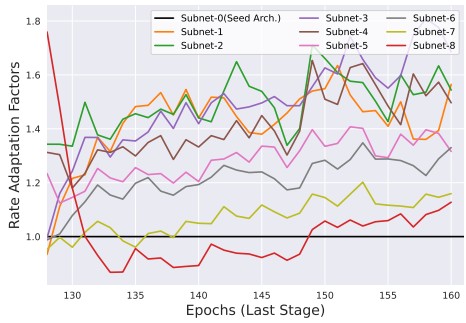

Figure 5: Visualization of our adaptive LR.

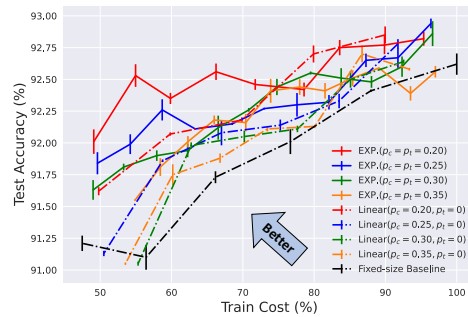

Figure 6: Comparison of growing schedules.

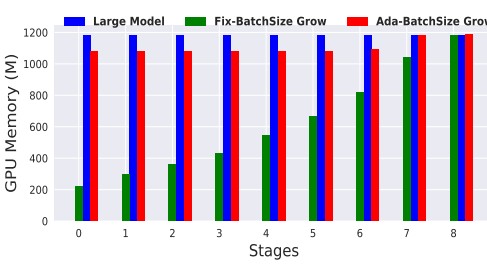

(a) ResNet-18 GPU memory allocations

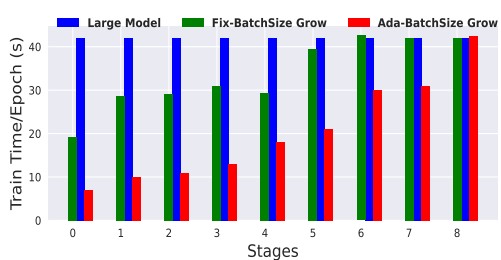

(b) ResNet-18 training time

Figure 7: Track of GPU memory allocations and wall clock training time for each growing phase.

**Flexible Growing Scheduler.** Our growing scheduler gains the flexibility to explore the best trade-offs between training budgets and test performance in a unified configuration scheme (Eq. 6 and Eq. 7). We compare the exponential epoch scheduler ($p_t \in \{0.2, 0.25, 0.3, 0.35\}$) to a linear one ($p_t = 0$) in ResNet-20 growing on CIFAR-10, denoted as 'Exp.' and 'Linear' baselines in Figure 6. Both baselines use the architectural schedulers with $p_c \in \{0.2, 0.25, 0.3, 0.35\}$, each generates trade-offs between train costs and test accuracy by alternating $T_0$. The exponential scheduler yields better overall trade-offs than the linear one with the same $p_c$. In addition to different growing schedulers, we also plot a baseline for fixed-size training with different models. Growing methods with both schedulers consistently outperforms the fixed-size baselines, demonstrating that the regularization effect of network growth benefits generalization performance.

**Wall-clock Training Speedup.** We benchmark GPU memory consumption and wall-clock training time on CIFAR-100 for each stage during training on single NVIDIA 2080Ti GPU. The large baseline ResNet-18 trains for 140 minutes to achieve 78.36% accuracy. As shown in the green bar of Figure 7(b), the growing method only achieves marginal wall-clock acceleration, under the same fixed batch size. As such, the growing ResNet-18 takes 120 minutes to achieve 78.12% accuracy. The low GPU utilization in the green bar in Figure 7(a) hinders FLOPs savings from translating into real-world training acceleration. In contrast, the red bar of Figure 7 shows that our batch size adaptation results in a large proportion of wall clock acceleration by filling the GPU memory, and corresponding parallel execution resources, while maintaining test accuracy. ResNet-18 trains for 84 minutes (1.7× speedup) and achieves 78.01% accuracy.

## 5 CONCLUSION

We propose an efficient and accurate method for network growing, based on principled rules regarding both parameterization and optimization. Our parameter transition from older to newer subnetworks is general and quick to execute when expanding the network. Our carefully designed learning rate adaptation mechanism improves optimization dynamics in networks consisting of subcomponents with heterogeneous training durations. Applications to widely-used architectures on image classification and machine translation tasks demonstrate that our method bests the accuracy of competitors, even outperforming the original fixed-size models, while saving considerable training cost.

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

# A  APPENDIX

Table 6: Rate Adaptation Rule for Adam (Kingma & Ba, 2015) and AvaGrad (Savarese et al., 2021).

| | | Our LR Adaptation |
|---|---|---|
| Adam | 0-th Stage | $\boldsymbol{m}_{\boldsymbol{t}[0]}\big/\sqrt{\boldsymbol{v}_{\boldsymbol{t}[0]}}$ |
| | i-th Stage | $\dfrac{\boldsymbol{m}_{\boldsymbol{t}[i]}\backslash\boldsymbol{m}_{\boldsymbol{t}[i-1]}}{\sqrt{\boldsymbol{v}_{\boldsymbol{t}[i]}\backslash\boldsymbol{v}_{\boldsymbol{t}[i-1]}}}$ |
| AvaGrad | 0-th Stage | $\dfrac{\boldsymbol{\eta}_{\boldsymbol{t}[0]}}{||\boldsymbol{\eta}_{\boldsymbol{t}[0]}/\sqrt{d_{\boldsymbol{t}[0]}}||_2}\odot\boldsymbol{m}_{\boldsymbol{t}[0]}$ |
| | i-th Stage | $\dfrac{\boldsymbol{\eta}_{\boldsymbol{t}[i]}\backslash\boldsymbol{\eta}_{\boldsymbol{t}[i-1]}}{||\boldsymbol{\eta}_{\boldsymbol{t}[i]}\backslash\boldsymbol{\eta}_{\boldsymbol{t}[i-1]}/\sqrt{d_{\boldsymbol{t}[i]}-d_{\boldsymbol{t}[i-1]}}||_2}\odot\left(\boldsymbol{m}_{\boldsymbol{t}[i]}\backslash\boldsymbol{m}_{\boldsymbol{t}[i-1]}\right)$ |

Table 7: Generalize to Adam and AvaGrad for ResNet-20 on CIFAR-10.

| Optimizer | Training Method | Preserve Opt. Buffer | Train Cost (%) | Test Acc. (%) ) |
|---|---|---|---|---|
| Adam | Large fixed-size | NA | 100 | 92.29 |
| Adam | Growing | No | 54.90 | 91.44 |
| Adam | Growing | Yes | 54.90 | 91.61 |
| Adam+our RA. | Growing | Yes | 54.90 | **92.13** |
| AvaGrad | Large fixed-size | NA | 100 | 92.45 |
| AvaGrad | Growing | No | 54.90 | 90.71 |
| AvaGrad | Growing | Yes | 54.90 | 91.27 |
| AvaGrad+our RA. | Growing | Yes | 54.90 | **91.72** |

## A.1  GENERALIZATION TO OTHER OPTIMIZERS

We generalize our LR adaptation rule to Adam (Kingma & Ba, 2015) and AvaGrad (Savarese et al., 2021) in Table 6. Both methods are adaptive optimizers where different heuristics are adopted to derive a parameter-wise learning rate strategy, which provides primitives that can be extended using our stage-wise adaptation principle for network growing. For example, vanilla Adam adapts the global learning rate with running estimates of the first moment $\boldsymbol{m}_t$ and the second moment $\boldsymbol{v}_t$ of the gradients, where the number of global training steps $t$ is an integer when training a fixed-size model. When growing networks, our learning rate adaptation instead considers a vector $\boldsymbol{t}$ which tracks each subcomponent's 'age' (i.e. number of steps it has been trained for). As such, for a newly grown subcomponent at a stage $i > 0$, $\boldsymbol{t}[i]$ starts as 0 and the learning rate is adapted from $\boldsymbol{m}_t/\sqrt{\boldsymbol{v}_t}$ (global) to $\frac{\boldsymbol{m}_{\boldsymbol{t}[i]}\backslash\boldsymbol{m}_{\boldsymbol{t}[i-1]}}{\sqrt{\boldsymbol{v}_{\boldsymbol{t}[i]}\backslash\boldsymbol{v}_{\boldsymbol{t}[i-1]}}}$ (stage-wise). Similarly, we also generalize our approach to AvaGrad by adopting $\boldsymbol{\eta}_t, d_t, \boldsymbol{m}_t$ of the original paper as a stage-wise variables.

**Preserving Optimizer State/Buffer** Essential to adaptive methods are training-time statistics (e.g. running averages $\boldsymbol{m}_t$ and $\boldsymbol{v}_t$ in Adam) which are stored as buffers and used to compute parameter-wise learning rates. Different from fixed-size models, parameter sets are expanded when growing networks, which in practice requires re-instantiating a new optimizer at each growth step. Given that our initialization scheme maintains functionality of the network, we are also able to preserve and inherit buffers from previous states, effectively maintaining the optimizer's state intact when adding new parameters. We investigate the effects of this state preservation experimentally.

**Results with Adam and AvaGrad** Table 7 shows the results growing ResNet-20 on CIFAR-10 with Adam and Avagrad. For the large, fixed-size baseline, we train Adam with $lr = 0.1, \epsilon = 0.1$ and AvaGrad with $lr = 0.5, \epsilon = 10.0$, which yields the best results for ResNet-20 following Savarese et al. (2021). We consider different settings for comparison, (1) optimizer without buffer preservation: the buffers are refreshed at each new growing phase (2) optimizer with buffer preservation: the buffer/state is inherited from the previous phase, hence being preserved at growth steps (3) optimizer with buffer and rate adaptation (RA): applies our rate adaptation strategy described in Table 6 while also preserving internal state/buffers. We observes that (1) consistently underperforms (2), which suggests that preserving the state/buffers in adaptive optimizers is crucial when growing

Table 8: Comparisons among Standard SGD, LARS and Ours for ResNet-20 Growth on CIFAR-10.

| Optimizer | Test Acc. (%) ) |
|---|---|
| Standard SGD | $91.95 \pm 0.09$ |
| SGD with Layer-wise Adapt.(LARS) | $91.32 \pm 0.11$ |
| Ours | $\mathbf{92.53 \pm 0.11}$ |

networks. (3) bests the other settings for both Adam and AvaGrad, indicating that our rate adaptation strategy generalizes effectively to Adam and AvaGrad for the growing scenario. Together, these also demonstrate that our method has the flexibility to incorporate different statistics that are tracked and used by distinct optimizers, where we take Adam and AvaGrad as examples. Finally, our proposed stage-wise rate adaptation strategy can be employed to virtually any optimizer.

**Comparison with Layer-wise Adaptive Optimizer** We also consider LARS (Ginsburg et al., 2018; You et al., 2020), a layer-wise adaptive variant of SGD, to compare different adaptation concepts: layer-wise versus layer + stage-wise (ours). Note that although LARS was originally designed for training with large batches, we adopt a batch size of 128 when growing ResNet-20 on CIFAR-10. We search the initial learning rate (LR) for LARS over $\{1e\text{-}3, 2e\text{-}3, 5e\text{-}3, 1e\text{-}2, 2e\text{-}2, 5e\text{-}2, 1e\text{-}1, 2e\text{-}1, 5e\text{-}1\}$ and observe that a value of $0.02$ yields the best results. We adopt the default initial learning rate of $0.1$ for both standard SGD and our method. As shown in Table 8, LARS underperforms both standard SGD and our adaptation strategy, suggesting that layer-wise learning rate adaptation by itself – i.e. without accounting for stage-wise discrepancies – is not sufficient for successful growing of networks.

## A.2    MORE VISUALIZATIONS ON RATE ADAPTATION

We show additional plots of stage-wise rate adaptation when growing a ResNet-20 on CIFAR-10. Figure 8 shows the of adaptation factors based on the LR of the seed architecture from 1st to 8th stages (the stage index starts at 0). We see an overall trend that for newly-added weights, its learning rate starts at $> 1\times$ of the base LR then quickly adapts to a relatively stable level. This demonstrates that our approach is able to efficiently and automatically adapt new weights to gradually and smoothly fade in throughout the current stage's optimization procedure.

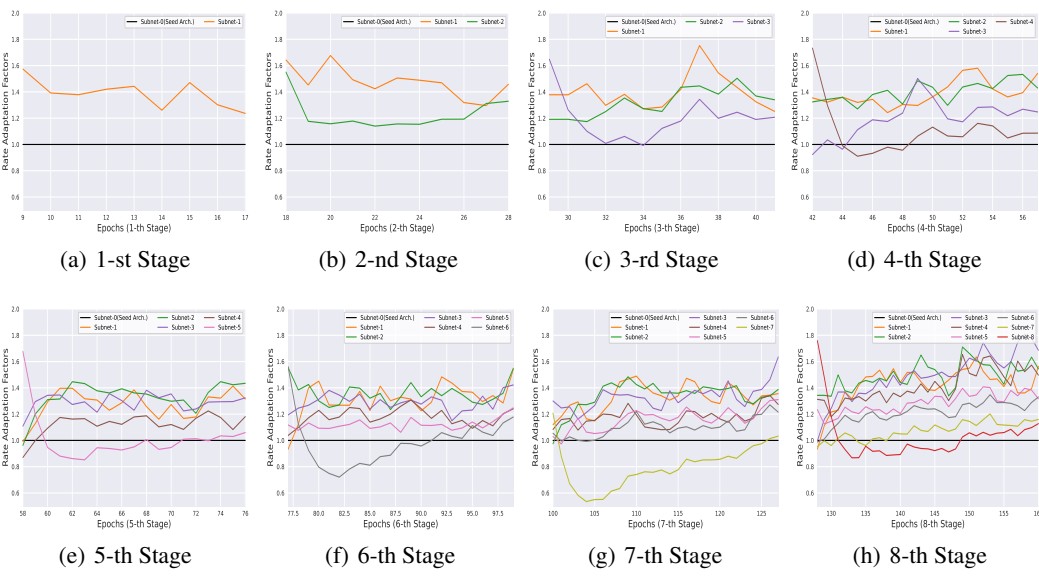

(a) 1-st Stage          (b) 2-nd Stage          (c) 3-rd Stage          (d) 4-th Stage

(e) 5-th Stage          (f) 6-th Stage          (g) 7-th Stage          (h) 8-th Stage

Figure 8: Visualization of Rate Adaptation Factor Dynamics across All Growing Stages (except 0-th)

### A.3 MORE VISUALIZATIONS ON SUB-COMPONENT GRADIENTS

We further compare global LR and our rate adaptation by showing additional visualizations of sub-component gradients of different layers and stages when growing ResNet-20 on CIFAR-10. We select the 2nd (layer1-block1-conv1) and 17th (layer3-block2-conv2) convolutional layers and plot the gradients of each sub-component at the 3rd and 5th growing stages, respectively, in Figures 9, 10, 11, 12. These demonstrate that our rate adaptation strategy is able to re-balance and stabilize the gradient's contribution of different subcomponents, hence improving the training dynamics compared to a global scheduler.

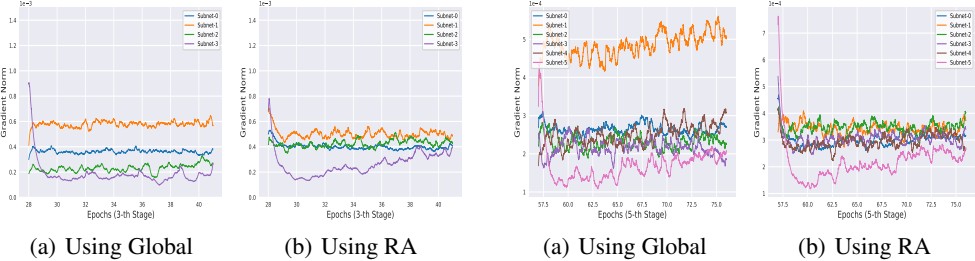

Figure 9: Gradients of 2nd conv at 3rd stage. Figure 10: Gradients of 2nd conv at 5th stage.

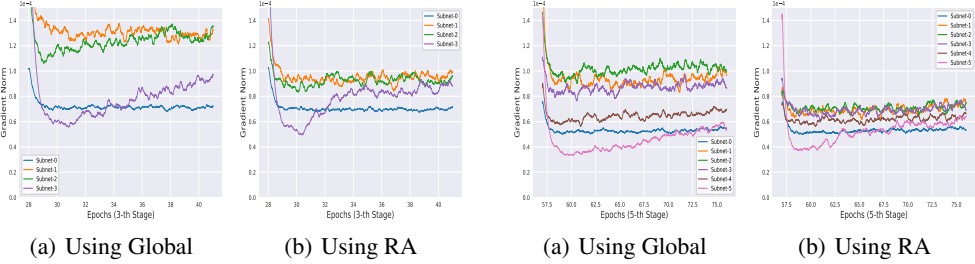

Figure 11: Gradients of 17th conv at 3rd stage. Figure 12: Gradients of 17th conv at 5th stage.

### A.4 SIMPLE EXAMPLE ON FULLY-CONNECTED NEURAL NETWORKS

Additionally, we train a simple fully-connected neural network with 8 hidden layers on CIFAR-10 – each hidden layer has 500 neurons and is followed by ReLU activations. The network is has a final linear layer with 10 neurons for classification. Note that each CIFAR-10 image is flattened to a

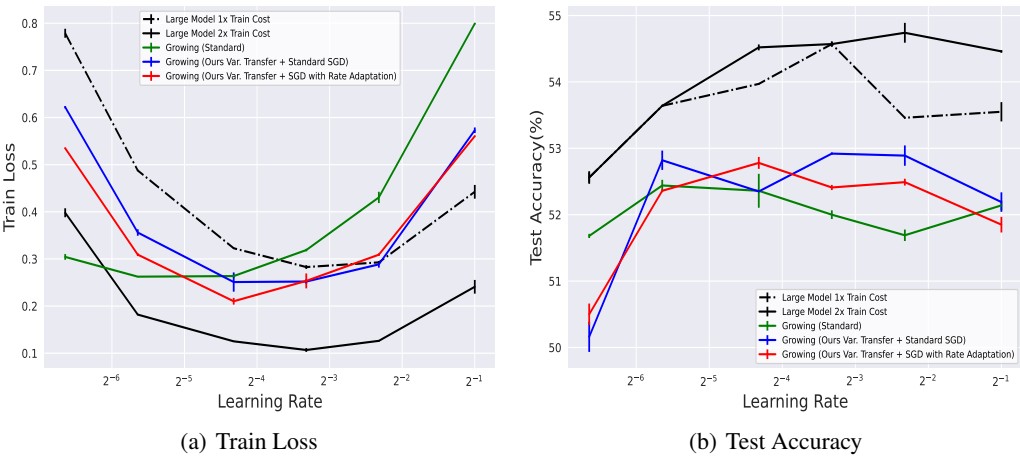

Figure 13: Results of Simple Fully-connected Neural Network.

3072-dimensional ($32 \times 32 \times 3$) vector as prior to being given as input to the network. We consider two variants of this baseline network by adopting training epochs (costs) $\in \{25(1\times), 50(2\times)\}$. We also grow from a thin architecture to the original one within 10 stages, each stage consisting of 5 epochs, where the number of units of each hidden layer grows from 50 to $100, 150, ..., 500$. The total training cost is equivalent to the fixed-size one trained for 25 epochs. We train all baselines using SGD, with weight decay set as 0 and learning rates sweeping over $\{0.01, 0.02, 0.05, 0.1, 0.2, 0.5\}$: results are shown in Figure 13(a). Compared to standard initialization (green), the loss curve given by growing variance transfer (blue) is more similar to the curve of the large baseline – all using standard SGD – which is also consistent with the observations when training model of different scales separately (Yang et al., 2021). Rate adaptation (in red) further lowers training loss. Interestingly, we observe in Figure 13(b) that the test accuracy behavior differs from the training loss one given in Figure 13(a), which may suggest that regularization is missing due to, for example, the lack of parameter-sharing schemes (like CNN) in this fully-connected network.

Table 9: Growing ResNet-18 using Incremental CIFAR-100.

| Method | Progressive Class | | Progressive Data | |
| --- | --- | --- | --- | --- |
| | Train Cost (%) $\downarrow$ | Test Accuracy (%) $\uparrow$ | Train Cost (%) $\downarrow$ | Test Accuracy (%) $\uparrow$ |
| Large fixed-size Model | 100 | 76.80 | 100 | 76.65 |
| Ours | 65.36 | 76.50 | 65.36 | 76.34 |
| Ours-Dynamic-OSGD | 68.49 | 77.53 | 68.49 | 77.85 |

## A.5 EXTENSION TO CONTINUOUSLY INCREMENTAL DATASTREAM

Another direct and intuitive application for our method is to fit continuously incremental datastream where $D_0 \subset D_1, ... \subset D_n... \subset D_{N-1}$. The network complexity scales up together with the data so that larger capacity can be trained on more data samples. Orthogonalized SGD (OSGD) (Wan et al., 2020) address the optimization difficulty in this context, which dynamically re-balances task-specific gradients via prioritizing the specific loss influence. We further extend our optimizer by introducing a dynamic variant of orthogonalized SGD, which progressively adjusts the priority of tasks on different subnets during network growth.

Suppose the data increases from $D_{n-1}$ to $D_n$, we first accumulate the old gradients $\boldsymbol{G_{n-1}}$ using one additional epoch on $D_{n-1}$ and then grow the network width. For each batch of $D_n$, we first project gradients of the new architecture ($n$-th stage), denoted as $\boldsymbol{G_n}$, onto the parameter subspace that is orthogonal to $\boldsymbol{G_{n-1}^{pad}}$, a zero-padded version of $\boldsymbol{G_{n-1}}$ with desirable shape. The final gradients $\boldsymbol{G_n^*}$ are then calculated by re-weighting the original $\boldsymbol{G_n}$ and its orthogonal counterparts:

$$\boldsymbol{G_n^*} = \boldsymbol{G_n} - \lambda * proj_{\boldsymbol{G_{n-1}^{pad}}}(\boldsymbol{G_n}), \quad \lambda : 1 \to 0 \tag{9}$$

where $\lambda$ is a dynamic hyperparameter which weights the original and orthogonal gradients. When $\lambda = 1$, subsequent outputs do not interfere with earlier directions of parameters updates. We then anneal $\lambda$ to 0 so that the newly-introduced data and subnetwork can smoothly fade in throughout the training procedure.

**Implementation Details.** We implement the task in two different settings, denoted as' progressive class' and 'progressive data' on CIFAR-100 dataset within 9 stages. In the progressive class setting, we first randomly select 20 classes in the first stage and then add 10 new classes at each growing stage. In the progressive data setting, we sequentially sample a fraction of the data with replacement for each stage, i.e. $20\%, 30\%, ..., 100\%$.

**ResNet-18 on Continuous CIFAR-100:** We evaluate our method on continuous datastreams by growing a ResNet-18 on CIFAR-100 and comparing the final test accuracies. As shown in Table 9, compared with the large baseline, our growing method achieves $1.53\times$ cost savings with a slight performance degradation in both settings. The dynamic OSGD variant outperforms the large baseline with $1.46\times$ acceleration, demonstrating that the new extension improves the optimization on continuous datastream through gradually re-balancing the task-specific gradients of dynamic networks.

