# OpenReview forum: "Accelerated Training via Principled Methods for Incrementally Growing Neural Networks"
_ICLR.cc/2023/Conference — Submitted to ICLR 2023_

### Official Review · Reviewer_3rDh · 2022-10-24

**Confidence:** 3
**Correctness:** 3
**Technical Novelty And Significance:** 2
**Empirical Novelty And Significance:** 3
**Recommendation:** 6

**Clarity, Quality, Novelty And Reproducibility:**

Clarity: This paper is mostly clear to read, however the description in section 3 feels cluttered with too much unnecessary notation. Nonetheless, Figure 2 is a good addition that conveys the message compactly.

Originality: The methods in this paper are moderately original. While the overall strategy does not deviate too broadly from those used in prior works (e.g.: Net2Net), the inspiration from the scaling used in the mutransfer paper makes the methods novel.

Quality: The paper scores moderate to low on quality. The methods come with little to no justifications, and the experiments have some ablations missing. Despite this, the large scale experiments on Imagenet are appreciated and show potentially that the proposed methods can scale to larger models.

Reproducibility: The method seems mostly reproducible from description, however, some details like that of optimizer resetting, are missing.


**Strength And Weaknesses:**

**Strengths**:

+ The paper tackles an important problem of accelerating training via incrementally growing neural networks, and a good solution to this problem can have huge implications for large-scale training of neural networks.

+ The heuristics proposed in this paper, i.e., variance transfer and rate adaptation seem like reasonable approaches, and the experimental results (esp those on Imagenet) agree with that observations.


**Weaknesses**:

**Proposed methods are heuristics but claimed otherwise; lacking justification**

- The title and several other parts of the paper make the claim that the proposed methods are principled. This seems to be because the methods are inspired from the scalings used in the mutransfer paper (Yang et al., 2021). However, it is worth stressing that a principled strategy in one setting does not immediately imply it being a principled strategy in another setting without appropriate justification, as I elaborate in the next two points. I thus request the authors to remove claims regarding the principled nature of proposed methods.

- The justification for variance transfer is "to make the weights of the old subnetwork compatible with the entire weight tensor parameterization", where an assumption is made that the old weights are drawn from a gaussian distribution with std = (1 / h_out^2). However, this is true only at initialization, and the distribution of weights can diverge in principle during training. Question: Why use the variance at initialization as opposed to the empirical variance at the time of growing the model? What is the precise justification for variance transfer in the first place?

- The learning rate adaptation method has no justification provided (even an intuitive one) and is purely a heuristic. Suggestion: Perhaps some intuition can be obtained by analyzing the dynamics of a simple one hidden layer MLP model?

- Missing detail: How are SGD momentum vectors handled at the time of growing a model? Are they re-initialized from scratch for each stage of re-training, or are they carried over from previous stages of optimization?

**Missing ablations on realistic models**

- While the analysis experiments in Section 4.4 are appreciated, it is important to have ablations presented for the more realistic Resnet-20 / Resnet-18 / Mobilenetv1 models used in Tables 1 and 2. In particular, it would be interesting to compare the following baselines - (1) Growing method + LR adaptation (LRA) + Variance transfer (VT) (aka the "full" method, already presented), (2) Method + LRA only, (3) Method + VT only, (4) Method only, which involves simply adding the new rows / columns to weights according to the prescribed schedule. In the absence of principled justification, these thorough ablations are necessary to evaluate the usefulness of the method.


**Summary Of The Paper:**

This paper proposes to train neural network models incrementally to reduce training time. To this end it proposes two heuristics, the first being variance transfer, which involves re-parameterizing weights to account for their variance, and second being rate adaptation, which adjust the learning rate of newly added weights different from the weights already present in the model. Experiments show that these strategies result in more accurate neural networks that are trained faster than existing methods for incremental training.

**Summary Of The Review:**

I am currently providing a recommendation of reject. Overall, this paper does not really propose any interesting technical insights or ideas, rather it consists of heuristics that nonetheless achieve SOTA results, making it borderline. However, I am willing to raise my score given that the authors are willing to show ablation experiments on realistic CIFAR10 / CIFAR100 models which will show the exact contribution of the various components of the method.

---- Update ----
Based on the additional experiments showed in the rebuttals, I am changing my recommendation to that of accept. My original review still largely stands, that the paper proposes a collection of heuristics that achieve SOTA results, making it still borderline. However, the strong results with ablation experiments indicate that these techniques may be useful to practitioners interested in this problem. Further, as I mention in my response, I hope that paper tempers down claims of being principled in an update of the draft.

---

> ### Author Response · Authors · 2022-11-18
> **response to Reviewer 3rDh**
>
> Thank you for the review and comments. We address your points individually below.
>
> **Q: Principle statement. (request the authors to remove claims regarding the principled nature of proposed methods.)**
>
> A: Please see our general reply above.
>
> **Q: Why use the variance at initialization as opposed to the empirical variance at the time of growing the model? What is the precise justification for variance transfer in the first place?**
>
> A: As mentioned in the general reply, variance transfer is motivated by the hyperparameter transfer scheme of Yang et al. [1].  Their scheme is developed with the goal of stabilizing the scale of gradients, weights, and activations, producing weight updates that maintain the original scale of the weights.  Making this assumption of stability, and extending their scheme to the case of dynamically growing network width yields the weight scaling and initialization rules shown in Table 1.  As discussed in the general reply, we could instead account for the actual weight values in a dynamic manner; we provide an ablation experiment for this alternative, but find that it does not provide an advantage.
>
> [1] Greg Yang, et al. Tuning large neural networks via zero-shot hyperparameter transfer, In NeruIPS, 2021.
>
> **Q: The learning rate adaptation method has no justification provided (even an intuitive one) and is purely a heuristic.**
>
> A: LARS [2] provides a layer-wise rate adaptation to rebalance LR for different layers.  Our rate adaption shares a similar concept but focuses on the growing problem by separating the effects of subnets with different start time.  We do provide the justification of rate adaptation by visualizing the gradients in Figure 4(b)(c) and Appendix A.3, demonstrating that RA is able to rebalance and stabilize the gradient contribution of different subcomponents.
>
> [2] Boris Ginsburg, Igor Gitman, and Yang You. Large batch training of convolutional networks with layer-wise adaptive rate scaling, 2018
>
> **Q: How are SGD momentum vectors handled at the time of growing a model (also the 'optimizer resetting')?**
>
> A: For SGD, we reinitialize the momentum buffer for each growing stage.  The following experiment shows that reusing the buffer does not improve the performance empirically on ResNet-20 growing experiments:
>
> | Ours  |  Ours-w-buffer |
> |---|---|
> | $92.53\pm0.11$  	|   $92.38\pm0.09$	|
>
> We also note that the optimizer buffer preservation is indeed crucial for adaptive optimizers (e.g., Adam, AvaGrad), as shown in Table 5 and Table 7.
>
> **Q: Missing ablations on realistic models.**
>
> A: We have shown some ablations for ResNet-20 growing on CIFAR-10 in Figure 4a.  We also conduct more component analysis as suggested.
>
> For ResNet-20 on CIFAR-10:
>
> |  Full|  Method +RA |   Method +VT|  Method |
> |---|---|---|---|
> |   $92.53\pm0.11$	|   $92.24\pm0.11$	| $92.00\pm 0.10$|$91.62\pm0.12$ 	|
>
>
> For ResNet-18 on CIFAR-100:
>
> |  Full| Method +RA|  Method +VT|  Method|
> |---|---|---|---|
> |   $78.12\pm0.15$|   $77.74\pm0.16$| $77.27\pm0.14$|$76.82\pm0.17$ |
>
> We see that, both RA and VT boost the baseline growing method.  Both components are designed and woven together for a high-level vision: efficiently growing networks while maintaining excellent generalization accuracy.  Our full method achieves best the test accuracy.

---

> > ### Comment · Reviewer_3rDh · 2022-11-26
> > **Response**
> >
> > Thank you for your response, and for the additional experiments.
> >
> > Regarding the principled nature of the proposals, my original comments still stand, and I still believe the paper cannot claim to propose "principled methods" for growing models. Note that not being principled is absolutely fine, as long as the paper is honest about its contributions. It is disappointing that the authors have chosen not to remove these claims, and I strongly recommend once again that they reconsider this decision in an update to the draft.
> >
> > The additional experiments done on realistic models are highly appreciated, and it is nice to see that all components of the proposed method are useful to obtain good performance. I will increase my score based on these results.

---

> > > ### Author Response · Authors · 2022-11-27
> > > **removing principled claim**
> > >
> > > Thank you for the comments and updated score.
> > >
> > > **Q: Principle statement. (request the authors to remove claims regarding the principled nature of proposed methods.)**
> > >
> > > A: We will revise our writing to refer to the specific technical methods rather than any overarching claim of 'principle'.  Our intention was to communicate motivation, not create a controversy over the degree to which 'principle' requires grounding in theory.  Updating the PDF is disabled for this phase of the discussion, but we will change the paper title to 'Accelerated Training via Incrementally Growing Neural Networks using Variance Transfer and Rate Adaptation', and revise the text accordingly.

---

### Official Review · Reviewer_ePbb · 2022-10-27

**Confidence:** 3
**Correctness:** 3
**Technical Novelty And Significance:** 3
**Empirical Novelty And Significance:** 2
**Recommendation:** 6

**Clarity, Quality, Novelty And Reproducibility:**

### Clarity and Quality

The writing is clear enough, but some critical points are less clear than they should be.  For instance, the method(s) herein are motivated in the introduction as:
>Our contribution is to do so in a manner that preserves the function computed by the model at each growth step (functional continuity) and offers stable training dynamics, while also saving compute by leveraging intermediate solutions. More specifically, we use partially trained subnetworks as scaffolding that accelerates training of newly added parameters, yielding greater overall efficiency than training a large static model from scratch.

Yet after reading the paper, apart from the gains in efficiency, it's unclear which of the other advances really help contribute to stability in training.

- Figure 3a has interesting information about the variability of the different growing methods, but this is obscured by the over-plotting of each method at the sampled learning rate points.  Could this be re-plotted with jitter?  Or as a dodged bar graph with whiskers (since the line segments connecting the distributions carry no information)?
- It’s hard to know what to take away from Figure 3b.  On the surface it looks like the red is most resistant to a larger learning rate, but this is likely because the growth method is adaptively adjusting the learning rate, as detailed in section 3.  The condition that seems to be missing is a large network with an adaptive learning rate
- In Figure 4a, it’s true that  the rate adaptation *seems* more stable, but this can be measured and reported to be more precise.  You could treat this trace as a time series and report the autocorrelation for each method

### Minor questions
Question in section 3.1: does the $\nu_{0}$ global learning rate get adjusted also?
Question in section 4 (Large Baselines via Fixed-size Training): Does the initial learning rate correspond to $\nu_{0}$?

### Comments

The problem of incrementally growing networks seems like it’s almost dual with the problem of pruning trained networks (or training networks);  A question for the authors is have they thought about what the literature on neural network pruning (especially LTH-flavoured work on finding subnetworks) has to say about the problem of growing networks?

Similarly, I wonder how the authors would say this work connects  with the aims of distillation?  There, a smaller model is trained to approximate the function of a larger model.  Here, a larger  model is adaptively grown in a way that preserves the function of the smaller model.

A final point is that figure 7 showcases some nice results for saving time  (and memory for fixed batch size grow), but for ResNet18 models, it’s a bit artificial, as this is a very small model that will take a maximum of one GPU to train anyhow. This would be much more convincing if it were measuring the time /epoch of a much larger model, like modern LLMs (e.g T5-3B), that take a very long time to train

**Strength And Weaknesses:**

### Strengths
- The authors have tried hard to present a more principled and stable method for growing networks from smaller to larger capacities, which is an important problem especially in large language models.
- Figure 5 and 6 are refreshing to see, as is much of section 4.4.  The authors are attempting to help us better understand how each different component of Algorithm 1 works to yield the final results in Tables 1-5.
- The authors do a good job summarizing and surveying the related work in network growing, and also explain the details of each component of their method in section 3 very clearly, using Figure 2 to help explain the growth stage layer-wise.

### Weaknesses

- The worth of some of the core contributions aren't very clearly established in the paper.  For instance, the value of the variance transfer parameterization achieves comparable performance with competing methods (some gains in table 5, but tables 1-3 show mostly marginal gains over Net2Net and Firefly).
- Also, the last claim is too broad.  The authors adaptive method bests the large model baseline for the VGG models.  The phrase "even outperforming the original fixed-size models" makes it seem as if this is often the case.
- In section 4.1, the paragraph describing table 2 is a nice result, but somewhat overstated.  On two models the authors' method outperform the baseline large model.  And on most comparisons (CIFAR 10/100), the gain is on par with Net2Net.  If the authors could find a test where their method was substantially better in test accuracy than Net2Net, it would really bolster their claims that Net2Net’s heuristic splitting method needs improving. Though the Transformer results in Table 5 display a somewhat more substantial difference in BLEU score.
- Is there a reason that distributions were reported in tables 1 through 3 but not tables 4 and 5?
- In the description of the results for Figure 3b, Blue is sometimes dominating orange in Figure 3b, but it’s not consistent especially at the lower learning rates (which are more consistent with what are used in practice)


**Summary Of The Paper:**

This paper considers the problem of growing neural networks during training.  The authors propose a parameterization and optimization scheme that pays close attention to weight and gradient scaling while reacting to training dynamics.  They further refine their original proposal to mitigate the problem of newly grafted subnetworks, that are relatively un-trained relative to the existing network, by an adaptive gradient scheme to ensure the contribution to the gradient is more balanced across new and older components.  They provide experimental evidence to show the efficacy of their method, paying particular attention to the savings in training time (or computation budget) accrued relative to larger monolithic networks.


**Summary Of The Review:**

While the authors have proposed what is likely an advance in model growth, they have been hampered somewhat by the choices they made in their experiments.

Incremental model growing is at its most useful when growing to very large models (think large language models circa 2021), )that would trouble even industry research labs.  Demonstrating a convincing win here in terms of performance versus compute (and time) saved would represent a more effective contribution, yet they focused much of their experiments on smaller vision models that do not offer a compelling reason to be grown stage-wise.

For this reason, I think that the paper offers some value, if rather marginal.

---

> ### Author Response · Authors · 2022-11-18
> **response to Reviewer ePbb**
>
> Thank you for the review and comments. We address your points individually below.
>
> **Q: The worth of some of the core contributions aren't very clearly established in the paper.**
>
> A: In addition to the component analysis in Figures 3, 4a and 13, we conduct a more comprehensive ablation study for ResNet-20 and ResNet-18.
>
> For ResNet-20 on CIFAR-10:
>
> |  Ours|  Baseline +RA |   Baseline +VT|  Baseline |
> |---|---|---|---|
> |   $92.53\pm0.11$	|   $92.24\pm0.11$	| $92.00\pm 0.10$|$91.62\pm0.12$ 	|
>
> For ResNet-18 on CIFAR-100:
>
> |  Ours|  Baseline +RA |   Baseline +VT|  Baseline |
> |---|---|---|---|
> |   $78.12\pm0.15$|   $77.74\pm0.16$	| $77.27\pm0.14$|$76.82\pm0.17$ 	|
>
> This shows that both RA and VT improve the growing baseline.  Our full method yields the best performance.
>
> **Q: Last claim too broad.**
>
> A: We will add the clarification that our method can outperform large fixed-size VGG baselines.
>
> **Q: In section 4.1, the paragraph describing table 2 is a nice result, but somewhat overstated.**
>
> A: We highlight the improvements over Net2Net on CIFAR-10 and CIFAR-100 by translating entries from Tables 2 and 3 as below.
>
> |  	|   ResNet-20|  VGG-11 |   MobileNetV1|
> |---|---|---|---|
> | Gains over Net2Net  	|  0.93 	|   0.56	|   1.67	|
>
>
> |  	|   ResNet-18|  VGG-19 |   MobileNetV1|
> |---|---|---|---|
> | Gains over Net2Net  	|  1.64 	|   1.38	|   1.52	|
>
> For CIFAR-10/100 image classification task, the above test accuracy improvements under the same training costs are sufficient to demonstrate that our method solidly outperforms Net2Net.
>
> **Q: Is there a reason that distributions (variance) were reported in tables 1 through 3 but not tables 4 and 5?**
>
> A: We did not show ImageNet result variance due to the heavy computation required.  We update with variance for the Transformer experiments below and will update ImageNet experiments in the final version.
>
> |   Large|  Net2Net |  Ours-w/o buffer|   Ours-w buffer| Ours-w buffer-RA|
> |---|---|---|---|---|
> |   $32.82 \pm 0.21$ |  $30.97\pm 0.35$	|  $31.44\pm0.18$	| $31.62\pm0.16$	|   $32.01\pm 0.16$|
>
>
> **Q: In the description of the results for Figure 3b, Blue is sometimes dominating orange in Figure 3b, but it’s not consistent.**
>
> A: Figure 3 shows an overall trend that variance transfer and rate adaptation provides a more stable loss/accuracy landscape across different learning rates.  We will delete the 'consistent' statement in the final version.
>
> **Q: It's unclear which of the other advances really help contribute to stability in training.**
>
> A: For (1) and (3), Figure 3 and 4(a) carry more information, such as generalization accuracy of different components, robustness across different learning rates.  In the final paper revision, we will extract specific information from these figures to highlight the stability property as suggested.
>
> For (2), our learning rate adaptation (RA) applies for subnets with different ages, which is specified for network growth (i.e., width changing during training).  It is not applicable to large fixed-size baseline training.  As a component analysis, Figure 3 is a fair comparison between our approach and large baselines to demonstrate the benefit of RA.
>
> Figure 4(b)(c) and Figure 9-12 seem overlooked, which are specifically designed to demonstrate the advance in training stability.  Those visualizations show that RA is able to re-balance and stabilize the gradient contribution of different subcomponents, hence improving the training dynamics compared to a global scheduler.

---

> > ### Author Response · Authors · 2022-11-18
> > **response to Reviewer ePbb (con't)**
> >
> > **Question in section 3.1: does the global learning rate $v\_0$ get adjusted also? Question in section 4 (Large Baselines via Fixed-size Training): Does the initial learning rate correspond to $v\_0$?**
> >
> > A: The global learning rate $v\_0$ follows the cosine learning rate scheduler and does not get adjusted by RA. The initial learning rate of large baselines correspond to $v\_0$.
> >
> > **Q: Connect to LTH.**
> >
> > A: As we state in related work section, we focus on the mechanics of growth when the target architecture is known.  Both LTH and pruning are orthogonal to and potentially compatible with our approach.  For example, even as we build the network, there could be a lottery ticket subset of each added stage whose union form a lottery ticket for the complete network.  Note that LTH concerns retraining; unlike our approach it does not currently provide a technique for accelerating training from scratch.
> >
> > **Q: Connect to model distillation.**
> >
> > A. Both our technique and model distillation involve the concepts of function preservation, but apply them for totally different purposes.  In model distillation, a small (student) model approximates the knowledge in a well-trained teacher network, and function preservation is the final goal of the optimization.  There is still significant expense in training the teacher network from scratch.  It is this from scratch training phase that we focus on accelerating; here, function preservation facilitates the knowledge transfer between one partially trained subnetwork and a larger network containing it.
> >
> > **Q: Super large model training.**
> >
> > A: This paper demonstrates the proof of concept with results up to ImageNet scale.  Yes, extremely large models are a target for future work; we will need partners with access to that level of compute resources.

---

> > > ### Comment · Reviewer_ePbb · 2022-11-24
> > > **Response to authors**
> > >
> > > >A: As we state in related work section, we focus on the mechanics of growth when the target architecture is known. Both LTH and pruning are orthogonal to and potentially compatible with our approach. For example, even as we build the network, there could be a lottery ticket subset of each added stage whose union form a lottery ticket for the complete network. Note that LTH concerns retraining; unlike our approach it does not currently provide a technique for accelerating training from scratch.
> > >
> > > I'm not sure I agree.  More broad versions of pruning where entire filters or units are pruned from the respective layers is not orthogonal to adaptively growing a network, it is more like a change in perspective.  While pruning is undertaken to maintain performance while improving efficiency (by discarding units that do not degrade overall accuracy), adaptively growing a network is undertaken to gain a marginal improvement in performance.  What I was suggesting is that there may be a deeper connection that the authors could explore, and that I for one would love to read :)
> > >
> > > I do appreciate the authors thoughts about how adaptive network growth relates to the concept of distillation, and thank them for attending to the other questions I raised.

---

> > > > ### Author Response · Authors · 2022-11-27
> > > > **Connect to more broad versions of pruning**
> > > >
> > > > Yes, broad versions of pruning (filter/layer pruning) relate to the growing concept in terms of dynamic architectures.  As we focus purely on growing, pruning is a separate and potentially compatible strategy that could be applied to our grown networks -- i.e., they could be subsequently or incrementally pruned.  From an algorithmic perspective, there might be modularity here, but you are correct that there could also be a deeper relationship if viewed jointly; we will revise our writing to clarify.
> > > >
> > > > Some existing prior work does try to combine growing and pruning for better training and inference efficiency.  For example, [1,2] start from seed architectures and adopt a growing and pruning scheme for architectural exploration and re-configuration.  Yuan et al. [2] need both growing and pruning aspects to achieve a ~2x training speedup.  We are able to do so with growing alone, suggesting advantages to our technical approach to growing.
> > > >
> > > > [1] Dai, Xiaoliang, Hongxu Yin, and Niraj K. Jha. ‘NeST: A neural network synthesis tool based on a grow-and-prune paradigm.’ IEEE Transactions on Computers, 2019.
> > > >
> > > > [2] Xin Yuan, Pedro Savarese, and Michael Maire. Growing efficient deep networks by structured continuous sparsification. In ICLR, 2021

---

> > ### Comment · Reviewer_ePbb · 2022-11-24
> > **Response to authors**
> >
> > Thanks for taking the time to consider my comments and post a response.  The additional ablation studies establish more clearly the contributions through subnet-specific learning rate adjustment and variance transfer.
> >
> > I think that the true value of this technique lies in how to expand a smaller instance of a much larger class of models (e.g LLMs) into much larger instances, and so I'm a bit disappointed that the authors were less enthusiastic about displaying their method's utility in this scenario.  I hope they will reconsider in the future.
> >
> > Beyond the additional results on ResNet-20 on CIFAR-10, the authors have only stated that they *will* make changes to the  final version of the manuscript, so my evaluation of the paper is unchanged for the present.

---

> > > ### Author Response · Authors · 2022-11-27
> > > **Super large model training**
> > >
> > > Thank you for the discussion.
> > >
> > > We are excited about the potential for such developments and would like to pursue super large model training.  However, we do not have the level of computational resources that would allow us to, say, run the experiment of T5-3B [1] LLM pre-training from scratch during the author response period.  While they do not quote the exact training FLOPs required, the training setup of [1] ran across TPU pods, each consisting of 1024 TPU v3 chips.  Our training speedups are in the ~2x regime, which, while significant, will still require substantial compute for large models.
> > >
> > > Our paper introduces the new ideas leading to this ~2x speedup, with experimental backing at the same scale as recently published prior work (e.g., Net2Net, FireFly).  Publishing our work can be a pathway to partnerships that enable super larger-scale experiments.
> > >
> > > [1] Raffel, Colin, et al. 'Exploring the limits of transfer learning with a unified text-to-text transformer.' JMLR, 2020.

---

### Official Review · Reviewer_qzqu · 2022-10-28

**Confidence:** 5
**Correctness:** 3
**Technical Novelty And Significance:** 2
**Empirical Novelty And Significance:** 2
**Recommendation:** 5

**Clarity, Quality, Novelty And Reproducibility:**

Overall, I don't think this paper is well written. The introduction is thorough in its coverage of related work, but lacks structure. It is not entirely clear what the distinction between the related work and introduction sections are. Section 3 is at times too dense because it tries too hard to formalize the method without providing an explanation or intuition for it.

The mathematical notation is messy: The weights are indexed by stage $i$ ($W_{i-1}$ becomes $W_i$), but their sizes are not ($H_{in}$ becomes $\widehat{H_{in}}$). Why use $H_{out}$ (where $H$ is for hidden) to denote the size of the output layer? $concat$ is not a mathematical operator. Multiplication is at times written as $\times$ and sometimes as $\ast$ (the latter is usually used for convolutions). Set notation is used on matrices, which doesn't really make sense. I also find the notation of $W_i \setminus W_{i-1}$ a bit convoluted. Additionally, some minor typographic things in the mathematical writing: I would prefer $x^{-1}$ instead of $1/x$, and super- and subscripts should be wrapped in `\mathrm`.

The figures are too busy. For example, I would use a figure like 1(b) that just shows the growing method that uses positive/negative weights, but leave out the variance transfer, learning rate scaling, noise, etc. Figure 2 could use colours, and shouldn't have all the formulae from the text repeated.

Are figures 4(b) and 4(c) plots of the gradient norms, or plots of the step norms (i.e., is this before or after being scaled with the learning rate)? I am assuming before (since figure 5 seems to suggest that the gradients for earlier subnets would actually be scaled up). Why are the gradients for the seed architecture weights not plotted?

What is the noise that is added to the new weights for symmetry breaking? How does the norm of this noise compare to the norm of the initialization?

Minor comment in the text: The title "stage-wise learning rate adaptation" made me think that the learning for each subnetwork is only changed once per stage. It was only when I saw figure 5 that I realized that the learning rate is actually changing throughout training.

All in all, I find the paper lacks clarity and some details are missing for reproduction. The paper introduces some novel techniques that could be interesting, but it fails to provide novel insights due to a lack of ablations and careful investigation of the underlying dynamics.

**Strength And Weaknesses:**

This paper has some interesting ideas: I like the way that the layers are grown. It feels similar to splitting (i.e., duplicating weights and then halving them) while avoiding changing the scale of the weights, which should help with the learning dynamics. The learning rate scaling is also interesting. One of the main problems with growing networks is that it can take a long time for the new capacity to be used. Perhaps this scaling of the learning rate helps with this somehow.

My main issue with this paper is that it proposes a lot of things (a function-preserving growing method, scaling of weights, learning rate scaling, growing schedules, batch size schedules, etc.). Many of these are not very theoretically grounded, and the ablations are limited. This makes it hard to disentangle which part of the methods are working and why. It gives a feeling of the authors having thrown the kitchen sink at the problem until something stuck.

The result is that this paper raises many questions for me:

* Is the variance transfer really useful? The effect seems minimal in figure 3. There are several learning rates for which the variance transfer and non-variance transfer curves overlap. Especially given that this is on CIFAR-10 with a 4-layer network, it seems like a bad idea to draw conclusions from these results. Without confidence intervals, figure 4 also is hard to draw definitive conclusions from (e.g., if training would have stopped at epoch 150 instead of 160, the methods would have been indistinguishable).
* Does variance transfer actually make sense? The scaling of the weights would only make sense if we know that the magnitude of the old weights hasn't changed much during training, but this isn't validated.
* What is the motivation for the learning rate scaling proposed in table 1? The intuition I would have is that you want to have a bigger learning rate for the new weights, for two reasons: they have to catch up, and the old weights can't be allowed to get too distorted by the initially random signals of the new weights. But the current learning rate scaling actually scales down the learning rate of the new weights when their norm is small, which seems like it might get them stuck? Would $\max(1, \lVert W_i \setminus W_{i-1} \rVert)$ maybe make more sense?
* Figure 4(c) shows that the gradient norms are closer together. In fact, the gradient norms of the oldest subnets seem to be lower, suggesting that they are better trained. But figure 5 shows that their weight norms (and hence learning rates) are still higher towards the end of training. This seems counterintuitive to me (usually you reduce the learning rate when you get closer to the optimum). And why is it that the weights of these old subnets stay so high compared to the initial weights? Is that because the variance transfer keeps scaling the old weights down?
* In figures 4(b), 4(c) and 5 you can see a very common dynamic of growing networks (look at subnets 7 and 8): When new weights are introduced they initially just produce noise, so the first thing the network does is to unlearn them (usually by driving them to zero). After that, the network takes some time to integrate this new capacity, at which point the gradient on these weights becomes similar to the rest of the network. My concern is that the proposed learning rate scheduling is slowing down this process by lowering the learning rate.

I could probably keep going and come up with more questions. (For example, I am wondering if on one hand the scheduling is helping by increasing the learning rate of older subnets compared to newer subnets, but on the other hand hurting by integrating new subnets slower in the network. Or maybe the lower learning rate is actually helping by giving the older network time to adapt to the newer weights before those settle?) My concern is that at the end, this paper provides a few interesting data points, but raises more questions than it answers. To argue for acceptance, I would like to see a paper with more focus (e.g., only the learning rate scheduling, or only the function-preserving growing method) and thoroughly study this through thoughtful ablations.

**Summary Of The Paper:**

This paper investigates methods to grow the width of linear neural network layers during training. Their proposed method consists of a variety of techniques:

* The new layer has a repeated block of new weights, which are multiplied by new submatrices in the next layer which only differ in sign, so that the final contribution of these weights is zero (or at least, almost).
* The old weights are scaled so that their expected variance is as if the weights were initialized using the fan-in of the new layer
* Throughout training, the learning rates for the new weights are scaled by the ratio of their norm and the norm of the original weights
* The layer sizes grow exponentially as a function of the growing stages
* The growing stages have an exponentially growing number of epochs
* The batch size decreases exponentially

The suggested method is compared on a variety of vision models (ResNet, VGG, MobileNet) and datasets (CIFAR-10, CIFAR-100, ImageNet) as well as a transformer trained for machine translation. They compare against several other methods (Net2Net, splitting, FireFly, GradMax). The results are encouraging, with the final test accuracy usually being very close to the large model trained from scratch while reducing training cost between 30% and 50%.

**Summary Of The Review:**

A paper that introduces some nice ideas, but it tries to do too many things and in the process fails to investigate the hard questions. The resulting paper feels like a long list of experiments and proposals without hypotheses that are tested. I would argue to reject this paper.

---

> ### Author Response · Authors · 2022-11-18
> **response to Reviewer qzqu**
>
> Thank you for the review and comments. We address your points individually below.
>
> **Q: It gives a feeling of the authors having thrown the kitchen sink at the problem until something stuck.**
>
> A: Please see our general response.
>
> **Q: Is the variance transfer really useful? The effect seems minimal in figure 3. Figure 4 also is hard to draw definitive conclusions from, when early stopping at 150 epochs.**
>
> A: Figure 3 illustrates that growing with variance transfer and rate adaptation provides a more stable loss/accuracy landscape across different learning rates.  In the 4-layer simple CNN example, variance transfer shows this property compared with standard initialization.  This property is more obvious when combining with rate adaptation.
> In Figure 4, one cannot directly compare the results by cutting the accuracy at 150 epochs due to the cosine annealing learning rate scheduler; when setting total training epochs to 150, the whole training process will change.  We train the model by setting the total training epochs as 150, and find VT still provides a performance gain over standard initialization:
>
> |    Standard   |      VT       |      VT+RA    |
> |:---:|:---:|:---:|
> |$91.56\pm0.11$|$91.89\pm0.12$|$92.42\pm0.11$|
>
> **Q: Does variance transfer actually make sense? The scaling of the weights would only make sense if we know that the magnitude of the old weights hasn't changed much during training, but this isn't validated.**
>
> A: Variance transfer is a general design and serves as a correction to Yang et al.'s parameterization [1], as that work does not consider training dynamics in which width grows incrementally.  One could alternatively involve the existing network's statistics through enforcing unit variance features, i.e., weight scaling is reformulated as $\sqrt{1/(fan\_{in} * var(W\_{old}))}$ instead, denoted as VT-constraint.  We compare with this variant by growing ResNet-20 on CIFAR-10. As shown in the table, both VT and VT-constraint outperform the standard baseline, which suggests standard initialization is a suboptimal design in network growing.  We also note that involving the weight statistics is not better than our simpler design.
>
> |  Standard|   VT|   VT-constraint|
> |:---:|:---:|:---:|
> |   $91.62\pm0.12$ 	|  $92.00\pm0.10$	| $91.93 \pm 0.12$ 	|
>
> [1] Greg Yang, et al. Tuning large neural networks via zero-shot hyperparameter transfer, In NeruIPS, 2021.
>
>
> **Q: What is the motivation for the learning rate scaling proposed in table 1.**
>
> A: The motivation of our LR adaptation is to compensate for discrepancy during growing by rescaling the LR for subcomponents.  Table 1 chooses weight norm statistics for SGD, providing an stage-wise extension to the layer-wise adaptation method LARS [2], i.e., $LR \propto ||W||$.  Note that one has the flexibility to utilize alternative adaptation rules (e.g., our Adam and AvaGrad extension in Appendix A.1).
>
> [2] Boris Ginsburg, Igor Gitman, and Yang You. Large batch training of convolutional networks with layer-wise adaptive rate scaling, 2018.
>
> **Q: The current learning rate scaling actually scales down the learning rate of the new weights when their norm is small, which seems like it might get them stuck. Would $max(1,||W\_i \setminus W\_{i-1}||)$ maybe make more sense?**
>
> A: Even if the norm of new weights is small, learning is unlike to get stuck.  Learning (weight updating) happens based upon effects of both gradients and learning rate adaptation.  As shown in Figure 4(c) and Appendix A.3, the gradients of the new subcomponent start with large values in first few epochs and then quickly adapt to a more stable level.
>
> Rate adaptation is a general design that different subnets should not share a global learning rate.  $max(1,||W\_i  \setminus W\_{i-1}||)$ is a plausible implementation choice, based on the assumption that new weights must have a higher learning rate.  We conducted experiments by growing ResNet-20 on CIFAR-10:
>
> |   Standard SGD|   Ours|   $max(1, \lVert W\_i  \setminus W\_{i-1} \rVert)$|
> |:---:|:---:|:---:|
> |   $91.62\pm 0.12$ |  $92.53\pm 0.11$|  $91.42\pm 0.12$ |
>
> We see that this alternative does not work better than our original design, and even underperforms standard SGD.

---

> > ### Author Response · Authors · 2022-11-18
> > **response to Reviewer qzqu (con't)**
> >
> > **Q: Figure 5 shows that their weight norms are counterintuitive.**
> >
> > A: Figure 5 plots only learning rate over time, not the weight norm.  Analyzing the learning process merely on weight norm (traversed from LR) is not reasonable.  In Figure 5 and Appendix A.2, there’s a clear trend for our dynamic approach that LR begins high for new subnets, which then quickly drops to accommodate the grown network training.
> >
> > **Q: When new weights are introduced they initially just produce noise, so the first thing the network does is to unlearn them (usually by driving them to zero); Concern is that the proposed learning rate scheduling is slowing down this process by lowering the learning rate.**
> >
> > A: This is not an accurate description of the training behavior.  As shown in visualizations of gradient norm and LR, the dynamic approach does provide large adjustments to new subnets, but this does not mean that weights are being driven to zero.  Our initialization scheme guarantees that the new weights, at random initialization, do not disrupt the function computed by the network -- as such, they cannot be viewed as injecting noise.
> >
> > **Q: Structure of Introduction.**
> >
> > A: The introduction presents a higher-level motivation and broader background of related work that suggests pathways for developing improvements in training efficiency.  The related work section narrows focus to specific prior work on which we build and competing techniques for growing networks that serve as baselines for comparison.
> >
> > **Q: Math notation and Figures.**
> >
> > A: We will revise them accordingly in the final version.
> >
> > **Q: Are figures 4(b) and 4(c) plots of the gradient norms, or plots of the step norms (i.e., is this before or after being scaled with the learning rate)? Why are the gradients for the seed architecture weights not plotted?**
> >
> > A: In the optimizer, gradient calculation (by back-propagation) and learning rate rescaling are decoupled and happen at each training step simultaneously.  Plotting these gradient norms is not dependent on LR adaption.  We do plot gradients for seed architecture in all figures, denoted as subnet-0.
> >
> > **Q: What is the noise that added to new weights for symmetry breaking.**
> >
> > A: Noise for symmetry breaking is 0.001 to the norm of the initialization.
> >
> > **Q: Stage-wise learning rate adaptation.**
> >
> > A: We will clarify that learning rate adapts along the whole training process in the method section.

---

### Author Response · Authors · 2022-11-18
**General Response**

We thank the reviewers, and answer specific questions in individual responses to each review below.

As recognized by Reviewers ePbb and 3rDH, we tackle an important problem, as a stable method for growing neural networks and accelerating training has potential widespread impact given the computational costs of large models.

Our empirical results are a dramatic improvement over the previous state-of-the-art, which is a fact that we believe some reviews missed.  Our gains over Net2Net in test accuracy (alternating columns of Tables 2, 3, 4, and 5) are anything but marginal.  In test accuracy, we improve over FireFly which itself improves over Net2Net, and are the sole approach with negligible difference in accuracy from the large model baseline.  Our method matches the speedup (training cost) of the fastest trainer (Net2Net), while producing trained models that are more accurate than all published approaches.

While we design many components to accomplish this empirical leap, in contrast to Reviewer qzqu's characterization, our approach is, in fact, the opposite of a 'kitchen sink': all technical components of our approach work together, and each addresses a separate aspect of stability when growing networks.

An analogy is the standard situation when training neural networks, where architectural components (e.g., residual connections), dynamic normalization schemes (e.g., batch norm), and optimization choices (e.g., learning rate schedule) interact.  There is a principle, though not a rigorously derived theorem, motivating the design of each (e.g., residual connections ameleorate gradient vanishing; batch norm reduces covariant shift).

When growing networks, we face an additional set of challenges in stabilizing the optimization, and invent a corresponding set of techniques to address these challenges.  The challenges are:

**Initialization:**
Network capacity is enlarged via adding new weights.  Naive random initialization of new weights destroys network functionality and may overwhelm any training progress.

**Weights Transition:**
Weight scaling from a previous growth step is not guaranteed to be maintained as the network architecture evolves.

**Optimization:**
Using a global learning rate schedule, different subnets will share the same learning rate, though they have trained for different durations.  One might expect this to give older, well-tuned, parameters too high a learning rate and newer parameters too low a learning rate; however, the gradient itself and other parameterization choices also influence the correct design for scaling weight updates.

---

> ### Author Response · Authors · 2022-11-18
> **General Response (con't)**
>
> Our corresponding technical solutions are:
>
> **Initialization with Functionality Preservation:**
> We preserve the function computed by the model at each growth step (functional continuity), which follows the principle of transferring previously learned knowledge and leveraging an intermediate solution as initialization for the next stage.  Additionally, newly initialized weights, though not impacting the computed function, are still nonzero and individually appear random, thereby facilitating learning.
>
> **Variance Transfer (VT):**
> We adopt the network parameterization scheme of Yang et al. (2021) for hyperparameter transfer and, assuming it suffices to stabilize weight magnitudes, extend it to the scenario of growing networks.  This yields our design for variance transition among networks of smaller and larger width in a single training process.
>
> As pointed out by reviewers, it is possible that our stability assumption, though itself a design goal of Yang et al.'s parameterization scheme, might be too strong.  A dynamic rescaling based on actual weight values could outperform our fixed rescaling based on relative network width.  To explore this, we provide an alternate VT formulation and conduct experiments on CIFAR-10 in response to Reviewer qzqu.
>
> We also conduct component analysis on CIFAR-10 and CIFAR-100 on various CNN networks to analyze variance transfer on realistic models in comparison to baselines, in response to Reviewers ePbb and 3rDH.
>
> **Learning Rate Adaptation:**
> Our learning rate adaptation (RA) scheme accounts for heterogeneous training durations of different subnets.  We provide justification by analyzing the gradient norms between a global learning and RA, where RA achieves behavior of rebalancing and stabilizing the gradient contribution of different subcomponents.  As additional experiments presented in replies below, we try the alternate formulation of RA as suggested by Reviewer qzqu; our original design is preferable, though has the flexibility to incorporate alternate rules.
>
> Each of these techniques can be viewed as `upgrading' an original part for training static networks into a corresponding one that accounts for dynamic growing.  There is a one-to-one mapping of these replacements and a guiding principle governing the formulation of each replacement.  Together, they accelerate training without impairing model accuracy -- a result that uniquely separates our approach from competitors.

---

### Decision · Program_Chairs · 2023-01-20

**Decision:**

Reject

**Justification For Why Not Higher Score:**

This paper could reasonably be accepted, but I lean towards rejection for the reasons outlined above.

**Justification For Why Not Lower Score:**

N/A

**Metareview: Summary, Strengths And Weaknesses:**

This paper presents a method for incrementally growing a neural network. The layers and weight matrices are gradually expanded, and the new weights are initialized by flipping the original weights. The main technical novelty consists of heuristics for rescaling parameters, adapting the learning rates, choosing the length of each training stage, and so on. On various image and language benchmarks, the proposed method is able to nearly match the test error of ordinary training while cutting the training cost by about a third.

The scores for the paper are somewhat borderline. Reviewers felt like the problem of incrementally growing networks is a useful one and the ideas seem fairly reasonable. On the other hand, they felt like the methods were somewhat of a bag of tricks with heuristic justifications, and the experiments didn't disentangle what was causing the improvement. In the revision, the authors introduced several ablation experiments which helped to address the latter criticism (and made some reviewers raise their scores). However, the contribution itself still feels incremental and heuristic. This is ok -- cleverness isn't a requirement in a paper -- but there needs to be enough analysis that readers will walk away having gained new insights. I think this could be served by finding cases where there are more substantial gains from incremental growing, or picking the tricks that seem most promising and subjecting them to deeper empirical and theoretical analysis.